# Natural Compounds as Modulators of Ferroptosis: Mechanistic Insights and Therapeutic Prospects in Breast Cancer

**DOI:** 10.3390/biom15091308

**Published:** 2025-09-11

**Authors:** Haotong He, Haoyang Yu, Hefeng Zhou, Guozhen Cui, Min Shao

**Affiliations:** 1Basic Teaching Department, Zhuhai Campus of Zunyi Medical University, Zhuhai 519041, China; hehaotong@zmuzh.edu.cn (H.H.); yuhaoyang@zmuzh.edu.cn (H.Y.); 2Department of Bioengineering, Zhuhai Campus of Zunyi Medical University, Zhuhai 519041, China; zhf@zmu.edu.cn (H.Z.); cuigz@zmu.edu.cn (G.C.)

**Keywords:** breast cancer, ferroptosis, natural compounds, drug resistance, ferroptosis inducers, phytochemicals, lead compounds

## Abstract

Breast cancer is the most prevalent malignant tumor in women. However, its clinical management is severely hindered by three interconnected challenges that limit long-term survival: treatment resistance, metastatic dissemination, and immunological evasion. Ferroptosis, an iron-dependent form of regulated cell death, is emerging as a novel strategy to overcome these obstacles. Furthermore, it demonstrates significant potential in inhibiting tumor metastasis and modifying the anti-tumor immune microenvironment, which provides a novel approach to address the core dilemma of breast cancer. Natural products have emerged as significant sources of ferroptosis inducers owing to their distinctive chemical variety, multi-target regulatory capabilities, and acceptable safety profile. Data increasingly indicates that several natural compounds can function as effective inducers or sensitizers of ferroptosis cell death. This review provides a thorough evaluation of current progress in harnessing natural ingredients to trigger ferroptosis for breast cancer treatment. It also elucidates the fundamental molecular mechanisms. Furthermore, it encapsulates therapeutic efficacy in preclinical models. Ultimately, it rigorously evaluates existing constraints and delineates potential and barriers for clinical translation.

## 1. Introduction

Breast cancer represents a major global health burden, continuing to be the most frequently diagnosed malignancy in women [1]. Despite ongoing advancements in treatment, clinical outcomes are frequently constrained by three critical challenges: acquired drug resistance, tumor metastasis, and immunological evasion [2,3,4]. These challenges jointly restrict long-term survival and persist as unresolved issues in contemporary oncology. Acquired resistance pervades the entirety course of chemotherapy, targeted therapy, and other conventional treatments, serving as a fundamental cause of therapeutic failure and disease recurrence. Distant metastasis is the principal cause of mortality associated with breast cancer [5,6]. Moreover, tumors create an immunosuppressive milieu that diminishes the effectiveness of immune-checkpoint inhibitors and other innovative immunotherapies [7]. Consequently, single-modality strategies infrequently attain lasting remission or cure. A comprehensive strategy that concurrently addresses resistance mechanisms, disrupts the metastatic process, and reconfigures antitumor immunity is essential for enhancing long-term survival and finally attaining clinical cure.

Ferroptosis is a controlled cell death mechanism characterized by iron buildup and lipid peroxidation, functioning independently of the classical apoptotic pathway [8]. Induction of ferroptosis can specifically eliminate drug-resistant cancer stem cells and metastatic precursor cells linked with epithelial–mesenchymal transition (EMT), hence inhibiting distant colonization [9]. During ferroptosis, damage-associated molecular patterns (DAMPs) are liberated. These signals stimulate dendritic cells and enhance the infiltration of CD8^+^ T lymphocytes, thus counteracting the immunosuppressive microenvironment [10,11]. The immunogenic cell can augment tumor sensitivity to immune-checkpoint inhibition. Consequently, the stimulation of ferroptosis is a multifaceted anticancer approach that targets three essential aspects: overcoming resistance, inhibiting metastasis, and modifying immunity.

Natural products exhibit exceptional structural diversity and broad biological activity, making them a significant resource for anticancer drug discovery [12]. Their multi-target and multi-pathway attributes correspond with the complex network that controls ferroptosis and may produce synergistic anticancer effects superior to those of single-target agents. Current natural-product libraries comprise a multitude of lead compounds. Notable compounds including curcumin and resveratrol have completed preclinical safety assessments, with some having progressed to clinical trials [13,14]. As a result, the systematic screening and validation of ferroptosis modulators generated from natural products have emerged as a leading area in drug development [15].

Although natural-product-based ferroptosis modulators exhibit intriguing promise in breast cancer treatment, existing research is predominantly confined to in vitro cell models and lacks comprehensive comparative data on sensitivity among various molecular subtypes [16]. Simultaneously, the multi-targeted regulatory mechanism of ferroptosis by natural compounds remains inadequately clarified and typically experiences low bioavailability and suboptimal pharmacokinetic properties [17]. The effective integration of ferroptosis inducers with established medicines, along with the absence of standardized ferroptosis assays and relevant biomarkers, has impeded their practical translation [18].

Investigating natural compounds that target the ferroptosis pathway has become as a crucial avenue for overcoming challenges in breast cancer treatment. This study summarizes recent advancements in modulating ferroptosis for breast cancer therapy. Specifically, it focuses on the role of natural products in this process. It emphasizes the fundamental molecular mechanisms. It aggregates preclinical research and assesses treatment efficacy in experimental models. Additionally, it offers a theoretical foundation for subsequent study and pharmaceutical innovation. Future research in this field is anticipated to focus on multiple areas. A key research priority is the development of biomarker-driven patient stratification systems to enable precision therapy. In parallel, it is essential to enhance the pharmacokinetic properties of these compounds through medicinal chemistry optimization and nanodelivery technologies. Equally important is a deeper analysis of the fundamental interplay between the tumor microenvironment and ferroptosis regulatory networks. It is essential to elucidate the interaction between metabolic reprogramming, immunological regulation, and ferroptosis. Furthermore, establishing standardized methodologies for the detection of ferroptosis is essential. It is also critical to develop dynamic monitoring techniques for ferroptosis-related biomarkers to facilitate clinical translation. These systematic investigations will provide a solid scientific basis and translational framework for novel therapeutic options targeting breast cancer.

## 2. The Molecular Regulatory Network of Ferroptosis

Ferroptosis is a type of programmed cell death that is driven by iron-dependent lipid peroxidation. The intricate regulatory network centers on three fundamental pillars, namely iron metabolism, lipid metabolism, and the antioxidant system [19]. A detailed elucidation of these molecular bases is crucial for comprehending how natural products exert anticancer effects through the targeting ferroptosis. This complex regulatory network, which visualizes the interplay between these core pillars, is schematically presented in Figure 1.

### 2.1. Regulation of Iron Metabolism

The onset of ferroptosis fundamentally contingent upon the abnormal enlargement the cellular labile iron pool (LIP). This enlargement results from impaired iron homeostasis, in which ferritinophagy is the primary pathway. During ferritinophagy, the adaptor protein Nuclear Receptor Coactivator 4 (NCOA4) is meticulously regulated by upstream signals, including Ataxia-telangiectasia mutated (ATM) kinase; in response to cellular stress, NCOA4 facilitates the lysosomal breakdown of ferritin complexes, liberating substantial quantities of catalytically active Fe^2+^ [20,21,22]. The released Fe^2+^ not only catalyzes Fenton reactions but also activates lipid-peroxidation enzymes, generating a self-amplifying positive feedback loop that ultimately leads to irreversible membrane damage and establishes a pro-ferroptotic intracellular milieu [23,24].

### 2.2. Lipid Peroxidation

At the molecular level, ferroptosis is defined by an oxidation cascade that specifically affects polyunsaturated fatty acids (PUFAs) in membrane phospholipids. The sensitivity of cells to ferroptosis correlates directly proportional to the quantity of oxidizable PUFAs in the membrane. Acyl-CoA synthetase long-chain family member 4 (ACSL4) functions as an initiating and specificity-determining enzyme by preferentially activating long-chain PUFAs such as arachidonic acid (AA) and adrenic acid (AdA) and facilitating their esterification into membrane phospholipids, thereby supplying specific substrates for subsequent lipid peroxidation [25,26]. In the presence of iron, these PUFAs are oxidized by lipoxygenases (ALOXs) and through non-enzymatic processes, resulting in the formation of cytotoxic lipid hydroperoxides (LOOHs) [27,28]. In response, cells increase the expression of stearoyl-CoA desaturase 1 (SCD1) to produce monounsaturated fatty acids (MUFAs), altering membrane composition from PUFA-rich to MUFA-enriched, thus establishing an inherent barrier against ferroptosis [29,30].

### 2.3. Antioxidant Defense System

To counter ferroptotic stress, cells have developed a multifaceted antioxidant system composed of parallel and partially redundant processes.

X4-dependent core pathway: The system cystine/glutamate antiporter (Xc^−^)–Glutathione(GSH)–Glutathione Peroxidase 4 (GPX4) axis constitutes the foundation. System Xc^−^, a heterodimer composed of Solute Carrier Family 7 Member 11 (SLC7A11) and Solute Carrier Family 3 Member 2 (SLC3A2), facilitates the transports extracellular cystine, which is then reduced intracellularly to cysteine within the cell and used for GSH biosynthesis. The rate-limiting step of GSH synthesis is catalyzed by glutamate cysteine ligase (GCLC). GSH is the essential cofactor for glutathione GPX4 in the reduction of lipid hydroperoxides [31,32]. Accordingly, the expression and activity of SLC7A11 directly affect baseline ferroptosis resistance [33,34].

X4-independent parallel pathways: Supplementary routes augment resilience. The Ferroptosis Suppressor Protein 1(FSP1)–Coenzyme Q10(CoQ10) pathway converts coenzyme Q10 to ubiquinol (CoQH_2_), a lipophilic antioxidant that scavenges lipid radicals [35,36,37,38]. The GCH1–BH4 route produces tetrahydrobiopterin (BH4) and associated metabolites, offering an additional lipid-antioxidant mechanism [19,39]. These pathways confer survival benefits when the canonical GPX4 system is impaired and exemplify a process of acquired resistance.

Upstream transcriptional regulation: Expression of these defenses is governed by a complicated transcriptional network. Nuclear factor erythroid 2 2-related factor 2 (NRF2), a principal regulator of oxidative-stress responses, enhances the expression of SLC7A11, GPX4, and ferritin, therefore inhibiting ferroptosis [40,41,42]. Conversely, p53 demonstrates bidirectional effects: wild-type p53 promotes ferroptosis by inhibiting SLC7A11 transcription, while certain mutant variants safeguard cancer cells, notably by augmenting NRF2 signaling—an effect observed in triple-negative breast cancer (TNBC) [43,44].

Ferroptosis is regulated by a dynamic network that equilibrates iron, lipid, and antioxidant systems balance each other. Cancer cells frequently modify or enhance their antioxidant defenses to endure ferroptosis; this adaptive metabolic reprogramming generates particular vulnerabilities that can be exploited therapeutically.

## 3. The Intrinsic Link Between Ferroptosis and Breast Cancer Therapy

### 3.1. Intrinsic Mechanisms Underlying Ferroptosis Sensitivity in Breast Cancer Cells

As a general principle in oncology, metabolic remodeling in cancer cells heightens reliance on iron and lipids, hence creating a foundational susceptibility to ferroptosis [45,46]. However, this sensitivity significantly differs due to subtype-specific molecular regulatory networks, primarily resulting from intrinsic differences among subtypes in lipid composition (e.g., the PUFA–MUFA balance) and the functionality of essential antioxidant pathways (e.g., the SLC7A11 and NRF2 signaling axes) [47,48,49]. Subtype-specific mechanisms are therefore crucial therapeutic targets because the ferroptosis regulation network is limited by intrinsic biological properties, such as transcriptional patterns and oncogenic signaling pathways. The key molecular pathways governing these subtype-specific responses are depicted schematically in Figure 2.

To support rapid proliferation, breast cancer cells typically demonstrate iron dependency via regulating iron metabolism proteins. Yu et al. indicated that in hypoxic conditions, HIF-1α enhances the expression of transferrin receptor 1 (TfR1), leading to elevated intracellular iron levels, the activation of ferroptosis, and the reversal of doxorubicin resistance [50]. Mobilizing intracellular iron stores is a crucial method for augmenting the labile iron pool (LIP). Huang et al. demonstrated that the natural product oridonin stimulates NCOA4-mediated ferritinophagy, hence increasing sensitivity to ferroptosis in triple-negative breast cancer (TNBC) [51]. Targeting iron-metabolism reprogramming is therefore an effective strategy to establish a pro-ferroptotic intracellular environment. Studies across various cancer types have established that the composition of membrane lipids is a fundamental factor influencing vulnerability to ferroptosis [52,53]. Polyunsaturated fatty acids (PUFAs), the precursors for lipid peroxidation, are activated and integrated into membrane phospholipids via a cascade involving acyl-CoA synthetase long-chain family member 4 (ACSL4) and Lys phosphatidylcholine acyltransferase 3 (LPCAT3). In breast cancer, ACSL4 expression is strongly correlated with ferroptosis sensitivity; in specific basal-like cell lines, ACSL4 promotes PUFA uptake and enhances susceptibility [53,54]. Xu et al. further established that thrombin causes ferroptosis in TNBC cells via the cPLA2α/ACSL4 axis [55]. LPCAT3, a crucial ferroptosis regulator, is transcriptionally controlled by factors like YAP and ZEB1, which are also key players in breast cancer biology [56].

### 3.2. Differences in Ferroptosis Sensitivity Across Molecular Subtypes of Breast Cancer

Various molecular subtypes of breast cancer have varying sensitivity to ferroptosis, attributable to distinct driver gene mutations, gene expression profiles, and reliance on core signaling pathways [15,48,57]. This inherent variability forms the theoretical basis for employing molecular classification to guide precision ferroptosis-targeted treatment.

TNBC typically demonstrates increased inherent sensitivity to ferroptosis inducers [58,59,60]. The molecular foundation of this hypersensitivity stems from the epithelial–mesenchymal transition (EMT) program, a process widely linked to ferroptosis sensitivity in multiple cancers [61,62,63], ZEB1 is a pivotal EMT transcription factor. Schwab and colleagues established that ZEB1 diminishes the production of MUFAs and facilitates the integration of PUFAs into membrane phospholipids, thereby increasing vulnerability to ferroptosis [64,65]. Kandettu et al. discovered in a separate investigation that mitochondrial microRNA-3 (mitomiR-3) targets and downregulates the ZEB1-mediated production of GPX4. This intervention remodels lipid metabolism, restructures lipid peroxidation, and finally induces ferroptosis in breast cancer cells [66]. The inherent connection between EMT and sensitivity to ferroptosis elucidates why the more aggressive TNBC subtype is paradoxically more susceptible to eradication by ferroptosis inducers.

Conversely, hormone-receptor-positive (HR-positive) luminal tumors typically exhibit heightened resistance to ferroptosis. This resistance is partially due to the subtype’s dependence on estrogen signaling and cell-cycle control pathways. For example, Cui et al. revealed that Cyclin-Dependent Kinase 4 and 6 (CDK4/6) inhibitors induce ferroptosis in Luminal A breast cancer cells by inhibiting SLC7A11. Nonetheless, the cells develop tolerance by compensatory upregulation of SLC7A11 [67]. Integrating ferroptosis inducers with drugs that inhibit fundamental HR-positive signaling pathways may thereby surmount this barrier.

The ferroptosis sensitivity of Human Epidermal Growth Factor Receptor 2 (HER2)-positive breast cancer is regulated by the HER2 signaling pathway and is generally low. Activation of HER2 initiates the phosphoinositide 3-kinase (PI3K)–AKT–mammalian target of rapamycin (mTOR) pathway, which promotes cell proliferation and concurrently bolsters lipid-oxidation defenses, thus inhibiting ferroptosis [68,69]. Nagpal et al. discovered Integrin αvβ3 as a principal regulator that safeguards HER2-positive tumors from ferroptosis induced by Tyrosine Kinase Inhibitors (TKI) [70]. Nonetheless, this established reliance on particular survival pathways presents possible treatment opportunities. Park and associates discovered that irreversible HER2 inhibitors can surmount resistance to the ferroptosis inducer RSL3 in non-HER2-amplified luminal breast cancer [71]. The data indicate that combination targeting of the HER2 pathway and ferroptosis mechanisms may offer an effective approach to overcome treatment resistance.

Although lipid hydroperoxides (LOOH) are the ultimate biochemical executioners of ferroptosis, their direct detection is highly challenging in breast cancer subtypes with intrinsic resistance to this cell death pathway [72]. This resistance primarily stems from an efficient antioxidant system that relies on glutathione (GSH) biosynthesis and its function as a cofactor for the enzymatic activity of glutathione peroxidase 4 (GPX4) [54]. The functional potency of this system has been well-validated experimentally. For instance, direct pharmacological inhibition of GPX4’s enzymatic activity with agents like RSL3, or depletion of GSH by blocking the upstream transporter SLC7A11 with inhibitors such as erastin, both lead to a sharp increase in intracellular LOOH levels in breast cancer cells [73,74]. This evidence strongly suggests that in many tumors with high expression of GPX4 and SLC7A11, such as certain luminal and HER2-positive subtypes, a dynamic equilibrium exists between LOOH production and clearance, keeping its steady-state concentration suppressed below detectable thresholds. Consequently, in these subtypes, the research focus naturally shifts from LOOH itself to the key upstream proteins that regulate its levels. A comprehensive summary of these differential sensitivities and their respective underlying mechanisms is presented in Table 1.

### 3.3. Genetic Regulatory Network of Ferroptosis Sensitivity

The vulnerability of breast cancer cells to ferroptosis is influenced by subtype-specific signaling pathways as well as a meticulously regulated network focused on essential genes such as p53, Breast Cancer vulnerability Gene 1 (BRCA1) and the PI3K–protein kinase B (AKT) pathway. The functional condition of these nodes directly determines how readily a cell undergoes ferroptosis.

In cells with wild-type p53, the protein mostly induces ferroptosis by directly inhibiting the transcription of SLC7A11, the essential subunit of the System Xc^−^ cystine–glutamate antiporter [77,78]. Huang and associates validated this hypothesis by demonstrating that the loss of the Farnesoid X Receptor (FXR) leads to p53 acetylation in a CREB Binding Protein (CBP)-dependent manner, consequently initiating ferroptosis and constraining breast cancer proliferation and metastasis [79]. By contrast, Dibra et al. revealed in a mouse model that mutant p53 efficiently protects TNBC cells from ferroptosis [44]. This discovery provides a molecular rationale for the treatment resistance typical of p53-mutant tumors. Thus, the mutation status of p53 is a significant biomarker for forecasting breast cancer’s susceptibility to ferroptosis inducers [80].

Constitutive activation of the PI3K/AKT pathway, commonly observed in breast cancer, constitutes a major mechanism that suppresses ferroptosis. Activated AKT phosphorylates and inhibits glycogen synthase kinase 3 beta (GSK3β), therefore alleviating the regulation of the transcription factor NRF2. Nuclear translocation of NRF2 subsequently induces the production of an antioxidant gene program, including GPX4, which jointly mitigates ferroptotic cell death [81,82]. Wu et al. established that loss of Phosphatase and Tensin Homolog (PTEN) mediates AKT activation, upregulates SLC7A11 through the GSK3β/NRF2 axis, and thereby suppresses ferroptosis in tumor cells [83].

Mutational inactivation of BRCA1 or BRCA2, characteristic lesions in hereditary breast cancer, is also closely associated with the regulation of ferroptosis. Lei et al. elucidated a multifaceted function of BRCA1; its loss causes cancer cells more susceptible to concurrent inhibition of GPX4 and Poly(ADP-ribose) Polymerase (PARP) [84]. Furthermore, Qu et al. indicated that, in cancers from patients with BRCA1/2 mutations, the expression and subcellular distribution of Ferritin Heavy Chain 1 (FTH1) are associated with risk of recurrence [85]. These data suggest that BRCA1/2 mutations may modulate ferroptosis sensitivity by disrupting iron-homeostasis pathways.

### 3.4. Strategies for Using Ferroptosis Induction to Overcome Drug Resistance in Breast Cancer

Inducing ferroptosis is a significant method for addressing drug resistance in breast cancer, as it operates through a molecular mechanism that is fundamentally distinct from classical apoptosis [86]. This distinction enables ferroptosis inducers to bypass established apoptosis-resistance pathways and provides novel therapeutic options for patients whose conditions advance following routine treatment. Ferroptotic cell death is independent of caspase activation. Consequently, it can eradicate cancer cells that evade apoptosis through over-expression of the anti-apoptotic protein B-cell lymphoma 2 (BCL-2) or through elevated levels of drug-efflux pumps such as P-glycoprotein [87,88]. Frye and associates established that even multidrug-resistant P-glycoprotein is incapable of efficiently inhibiting ferroptosis inducers [89]. Based on this result, Zhang et al. developed a novel taxoid, SB-T-101141. The drug disrupts the iron-homeostasis regulator KH-type Splicing Regulatory Protein (KHSRP), triggers an unconventional ferroptotic cascade, and overcomes paclitaxel resistance in breast cancer [90].

The anticancer efficacy of the traditional chemotherapeutic agent doxorubicin is somewhat facilitated by ferroptosis, resulting from the interplay of impaired iron metabolism and reactive oxygen species generation [91,92]. Cancer cells can, nevertheless, acquire adaptive resistance. Shen et al. showed that DnaJ Heat Shock Protein Family Member C12 (DNAJC12) stimulates the AKT pathway and inhibits doxorubicin-induced ferroptosis [93]. Zhu et al. indicated that GATA Binding Protein 3 (GATA3) confers resistance by obstructing Cytochrome B5 Reductase 2 (CYB5R2)-dependent iron reduction [94]. These findings delineate explicit targets for combination therapy. Inhibitors directed against DNAJC12 or GATA3 could be paired with doxorubicin to enhance its effectiveness [95].

The stimulation of ferroptosis holds significant potential for surmounting resistance to targeted therapy. Hua et al. discovered that trastuzumab-resistant HER2-positive breast cancer cells can be resensitized by the inhibition of SLC7A11-mediated cystine metabolism [96]. In estrogen-receptor-positive tumor, cells that acquire resistance to CDK4/6 inhibitors or tamoxifen are also vulnerable to ferroptosis [97,98,99]. Adaptive reprogramming in resistant cells enhances antioxidant defenses and establishes a crucial reliance on GPX4. This dependence delineates a therapeutic window [84,100,101]. Direct suppression of GPX4 or reduction of its cofactor glutathione can therefore exploit the metabolic vulnerability resulting from adaptive reprogramming [102,103].

Furthermore, ferroptosis is now understood to operate as a form of immunogenic cell-death (ICD) in the broader context of cancer immunology, transforming an immunosuppressive tumor microenvironment into an immune-active state [104,105]. Yang et al. noted that variability in ferroptosis potential within TNBC is associated with responsiveness to immunotherapy. Augmenting ferroptosis improved the effectiveness of anti-Programmed Death-1/Programmed Death-Ligand 1 (PD-1/PD-L1) checkpoint blockade [3]. However, Tumor cells can evade treatment by inhibiting ferroptosis. Luo and associates demonstrated that High Mobility Group Box 3 (HMGB3) suppresses interferon-γ-driven ferroptosis in TNBC and thereby promotes resistance to anti-PD-1 therapy [106]. The combination of ferroptosis inducers with immune checkpoint inhibitors has emerged as a promising synergistic anticancer strategy [107,108].

## 4. Natural Products as Modulators of Ferroptosis in Breast Cancer

As a distinct form of non-apoptotic regulated cell death, ferroptosis offers a novel strategy to overcome key challenges in breast cancer, as its molecular mechanism can effectively bypass obstacles such as therapeutic resistance and immune evasion. The complex molecular network governing ferroptosis, in turn, provides a wealth of molecular targets for precision therapeutic intervention. Among various strategies, natural products have garnered significant attention as a class of highly promising modulators. Their value lies primarily in their role as a source of established drug lead compounds, often possessing favorable safety profiles and inherent multi-target regulatory capabilities. Therefore, this chapter will systematically review the natural compounds reported to date that can induce or sensitize breast cancer cells to ferroptosis.

However, before delving into the specific biological activities of these various natural compounds, it is crucial to first acknowledge a prevalent and critical challenge: their common pharmacokinetic deficiencies. Many promising natural products, including highly-studied molecules like curcumin and resveratrol, are hampered by issues such as low aqueous solubility, poor intestinal absorption, and rapid in vivo metabolism. These factors collectively lead to low bioavailability, making it extremely difficult to achieve and maintain effective therapeutic concentrations at the tumor site. Therefore, understanding this fundamental bottleneck is essential for objectively evaluating the preclinical data discussed hereafter and for recognizing the necessity of developing optimization strategies, such as advanced delivery systems.

### 4.1. Polyphenols

The following discussion focuses on *Polyphenols* to modulate ferroptosis. A detailed overview of pivotal studies, outlining the compounds, their targets, and outcomes in breast cancer models, is provided in Table 2. The corresponding molecular structures are illustrated in Figure 3 for clarity.

Polyphenolic compounds comprise aromatic rings abundant in phenolic hydroxyl groups. This configuration allows for the chelation of Fe^2+^ and involvement in redox reactions. These two features constitute the chemical foundation for polyphenol-mediated ferroptosis [109,110]. In breast cancer research, these phytochemicals are attracting attention due to their ability to address ferroptosis [111,112]. The ferroptosis-modulating polyphenols are categorized into flavonoids and non-flavonoids based on their core skeletons. Recent investigations indicate that these chemicals elicit ferroptosis in breast cancer cells via several routes. They disturb iron homeostasis, directly impede antioxidant defense, and alter lipid metabolism.

Quercetin, a representative flavonoid, induces ferritinophagy, thus initiating ferroptosis. An et al. initially demonstrated that quercetin facilitates the nuclear translocation of Transcription Factor EB (TFEB). This event triggers lysosomal breakdown of ferritin, liberates stored iron, and induces ferroptosis in breast cancer cells [113]. However, quercetin can produce contrary effects in different settings. Zhu et al. demonstrated that it inhibits neuronal ferroptosis via modulating the lipid-metabolism gene Prostaglandin-Endoperoxide Synthase 2 (PTGS2), hence mitigating chemotherapy-associated cognitive impairment [114]. Despite the observation of this environment-dependent bidirectional regulation, the exact molecular switch responsible for such selection remains undetermined [115].

Resveratrol provides molecular evidence for disruption of the GPX4 pathway. Zhang et al. revealed that resveratrol attracts Neural Precursor Cell Expressed Developmentally Down-Regulated Protein 4-Like (NEDD4L) in TNBC cells. This E3 ubiquitin ligase facilitates the ubiquitination and degradation of GPX4, promotes ferroptosis, and suppresses tumor development in vivo [116]. This mechanism of action contrasts with that of the traditional inducer RSL3, which directly inhibits GPX4 enzymatic activity. The mechanism of resveratrol may therefore surmount resistance mediated by GPX4 overexpression or active-site mutation [116]. Other polyphenols attain comparable outcomes by modulating upstream processes. Hu et al. discovered that oxyresveratrol suppresses the Epidermal Growth Factor Receptor (EGFR)/PI3K/AKT pathway, hence reducing GPX4 expression [81].

Polyphenols additionally regulate lipid metabolism and mitochondrial activity. Formononetin prompts ferroptosis in TNBC cells by reprogramming lipid metabolism via the mechanistic target of rapamycin complex 1 (mTORC1)/Sterol Regulatory Element-Binding Protein 1 (SREBP1)/Stearoyl-CoA Desaturase 1 (SCD1) pathway [117]. Jiang et al. indicated that rosmarinic acid facilitates mitochondrial fission, thereby inducing ferroptosis in TNBC [118]. Research on bioflavonoids and the isoflavones genistein and daidzein further substantiates the perspective that flavonoids exert anticancer effects by targeting various ferroptosis-related nodes [119,120].

Curcumin exemplifies multi-target synergy in the activation of ferroptosis [121]. Hao et al. revealed that curcumin inhibits glutathione synthesis by obstructing the p53/SLC7A11 axis, hence undermining the fundamental antioxidant defense of cancer cells [122]. Zhang et al. subsequently established that curcumin amplifies metabolic stress by facilitating Solute Carrier Family 1 Member 5 (SLC1A5)-mediated glutamine absorption, thereby intensifying the ferroptotic effect [123]. Independent research revealed that curcumin elevates the labile iron pool by upregulating Heme Oxygenase-1 (HO-1). The additional Fe^2+^ provides the Fenton reaction with a catalytic substrate [124,125]. The simultaneous reduction of antioxidant defenses, the escalation of metabolic stress, and the accumulation of pro-oxidant iron collectively form an efficient strategy for inducing ferroptosis. This mechanism elucidates the extensive antitumor efficacy observed in diverse breast cancer models.

Numerous polyphenols demonstrate robust effectiveness in TNBC models, implying an inherent susceptibility to ferroptosis. Evidence indicates two primary factors: first, certain TNBC cells have elevated amounts of acyl-CoA synthetase long-chain family member 4 (ACSL4), the enzyme responsible for esterifying polyunsaturated fatty acids into membranes—an essential precursor for ferroptosis. Second, the dependence of TNBC on specific metabolic pathways, such as cystine metabolism, renders it more susceptible to drugs that inhibit GSH synthesis [112,126].

Besides serving as individual active molecules, polyphenols can function as chemosensitizers that mitigate resistance. Numerous studies indicate that curcumin enhances the sensitivity of breast cancer cells to standard chemotherapeutics like cisplatin or doxorubicin by cooperatively triggering ferroptosis [127,128]. Silybin modifies the tumor immune milieu in TNBC and augments antitumor immunity, thereby offering an immunological adjunct to ferroptosis-oriented therapy [129]. This combination method may mitigate the toxicity of cytotoxic agents and presents a novel option for treatment-resistant breast cancer.

Nonetheless, the clinical application of polyphenols encounters substantial obstacles, with generally inadequate bioavailability as the primary bottleneck. This limitation arises primarily from inherently low aqueous solubility and pronounced first-pass hepatic–intestinal metabolism following oral administration, which includes rapid enzyme-mediated conjugation activities (e.g., glucuronidation and sulfation) in the gastrointestinal tract and liver [130,131,132]. Researchers are exploring nanotechnology-based delivery techniques, including metal–polyphenol networks and biomimetic nanocarriers to solve this issue. These systems can safeguard active ingredients from premature degradation in circulation and exploit the tumor-specific enhanced permeability and retention (EPR) effect to facilitate passive accumulation, hence offering practical methods to boost bioavailability and tumor targeting [133,134].

**Table 2 biomolecules-15-01308-t002:** Polyphenolic compounds that induce ferroptosis in breast cancer.

Compound	Strength of Evidence	Chemical Class	Model/ Disease	Effect	Reference
Quercetin	Strong	Flavonoid	MCF-7, MDA-MB-231 (breast cancer)	Induces ferroptosis by driving TFEB nuclear translocation, activating ferritinophagy, expanding the labile iron pool and intensifying lipid peroxidation.	[113]
Robustaflavone A	Strong	Flavonoid (biflavonoids)	Breast cancer cell lines	Elicit ferroptosis through mitochondrial dysfunction–associated escalation of lipid peroxidation.	[119]
Genistein	Strong	Isoflavone	MDA-MB-231 (TNBC)	Triggers ferroptosis characterized by increased lipid ROS and MDA with concomitant GPX4 suppression.	[120]
Daidzein	Strong	Isoflavone	MDA-MB-231 (TNBC)	Promotes ferroptosis via enhanced lipid peroxidation and GPX4 downregulation.	[120]
Resveratrol	Strong	Polyphenol (stilbene)	TNBC cells and xenografts	Induces ferroptosis through NEDD4L-mediated ubiquitination and degradation of GPX4, leading to lipid peroxide accumulation.	[116]
Oxyresveratrol	Strong	Polyphenol (stilbene)	Breast cancer cells	Suppresses the EGFR/PI3K/AKT pathway, downregulates GPX4, and thereby provokes ferroptosis.	[81]
Formononetin	Strong	Isoflavone	TNBC cells	Inhibits the mTORC1/SREBP1/SCD1 axis, reduces MUFA biosynthesis, and facilitates ferroptosis via augmented lipid peroxidation.	[117]
Rosmarinic acid	Strong	Phenolic acid	TNBC cells	Promotes mitochondrial fission and consequently drives ferroptosis through intensified lipid oxidative damage.	[118]
Curcumin	Strong	Curcuminoid	MCF-7, MDA-MB-231 (breast cancer)	Orchestrates ferroptosis by repressing the p53/SLC7A11 axis to deplete GSH and inactivate GPX4, while upregulating HO-1 to enlarge the labile iron pool and enhancing SLC1A5-dependent metabolic stress.	[121]

### 4.2. Terpenoids and Their Derivatives

This section reviews the role of *Terpenoids* and Their Derivatives in modulating ferroptosis in breast cancer. A comprehensive summary of key findings, including specific compounds and their reported mechanisms, is presented in Table 3. For visual reference, the chemical structures of representative molecules discussed herein are displayed in Figure 4.

Terpenoids are structurally varied natural compounds, and their ability to cause ferroptosis in breast cancer frequently stems from electrophilic moieties within the framework. Specifically, α,β-unsaturated carbonyls function as Michael acceptors that covalently modify catalytic cysteines in targets such as GPX4, resulting irreversible inactivation and therefore facilitating ferroptosis [136,137]. In addition to GPX4, these electrophilic groups may also engage with other critical regulators of the ferroptotic pathway, thereby significantly influencing bioactivity [138].

Artemisinin and its derivatives are the most well researched examples; its distinctive endoperoxide bridge (-O-O-) confers antitumor activity against neoplastic cells at the molecular level. In Fe^2^-rich cellular environments, this bridge experiences selective cleavage, producing substantial quantities of reactive oxygen species (ROS)—a pivotal initiator of ferroptosis [139]. Subsequently, these ROS trigger a cascade of lipid peroxidation in PUFA-rich phospholipids, with the resultant accumulation of phospholipid hydroperoxides (PLOOH) considered the final stage of ferroptosis [140]. Researchers have developed techniques to improve therapy efficacy in breast cancer based on this mechanism. One method utilizes metal–organic frameworks (MOFs) as carriers that co-encapsulate artemisinin and a tannic acid–Fe (II) complex, concurrently delivering the drug and providing Fe^2+^ to tumor cells, thereby improving efficacy in TNBC models [141]. Chemically synthesized artemisinin hybrids and polymeric variants like poly(ARTEMA) have demonstrated the ability to cause both ferroptosis and apoptosis in MCF-7 breast cancer cells [142,143].

Subsequent research indicates that artemisinin derivatives operate via multiple apoptotic pathways. Dihydroartemisinin (DHA) increases radiosensitivity of breast-cancer cells by targeting circRNA hsa_circ_0001610 to regulate ferroptosis-related signaling [144]; DHA also activates the AIM2/Caspase-3/DFNA5 axis to produce pyroptosis [145]. These findings suggest that specific terpenoids can activate numerous programmed cell death pathways, providing a strategy to circumvent resistance resulting from the inactivation of any single route [140].

Mechanistic studies on additional terpenoids within the GSH/GPX4 regulation axis highlight complex regulatory mechanisms. A retrospective clinical study noted enhanced survival in breast cancer patients treated with the traditional Chinese medicine Danshen; subsequent research identified dihydroisotanshinone I as an inducer of ferroptosis and apoptosis. At the molecular level, tanshinone IIA controls KDM1A and thereby influencing PIAS4-mediated SUMOylation of SLC7A11, resulting in destabilization and degradation of SLC7A11 [146]. In resistant TNBC, curcumol suppresses the SLC7A11/NF-κB/TGF-β pathway, but its derivative HCL-23 kills cancer cells death through HO-1-dependent ferroptosis [135,147]. The data collectively indicate that terpenoids can provoke ferroptosis via affecting proteins upstream of GSH formation (e.g., SLC7A11) or by altering iron-metabolism enzymes such as HO-1.

Terpenoids utilize various ways to downregulate GPX4 through post-translational changes. Anomanolide C drives GPX4 into an autophagy-dependent degradation route via ubiquitination [148]. Taraxerol decreases NRF2 transcriptional activity, hence indirectly facilitating MIB2-mediated GPX4 ubiquitination and degradation [149]. Certain terpenoids influence upstream transcription factors: eupaformosanin demonstrates anticancer properties by ubiquitinating mutant p53 [150], whereas ursolic acid promotes ferroptosis in breast-cancer stem cells through the modulation of NRF2 [151]. The discovery regarding NRF2, typically a primary inhibitor of ferroptosis, indicates a complexity of context specificity complexity that necessitates additional examination.

The benefits of terpenoid molecules arise from the interaction of their chemical properties and the synergistic mechanisms they facilitate. Hinokitiol, for instance, forms compounds with iron and functions directly as an iron ionophore/carrier, thus circumventing intricate upstream regulatory checkpoints [152]. Other drugs demonstrate potentiation: oridonin amplifies the cytotoxicity of the GPX4 inhibitor RSL3 through the JNK/NRF2/HO-1 pathway [153]; cucurbitacin B, in conjunction with erastin, modifies iron-regulatory-protein expression and intensifies lipid peroxidation to collaboratively induce ferroptosis [154]. Many terpenoids structurally possess α, β-unsaturated carbonyl groups that act as Michael acceptors. These groups can establish covalent connections with thiol groups of critical cysteine residues in GPX4 or upstream regulators like Keap1 [155]. For instance, a study in esophageal cancer cells has demonstrated the covalent interaction between oridonin and GSH [156]. Covalent modification resulting in irreversible inhibition may provide more sustained target inactivation compared to reversible interactions, establishing the chemical foundation for terpenoids as lead molecules for next-generation ferroptosis-targeted anticancer treatments [157].

**Table 3 biomolecules-15-01308-t003:** Terpenoids and their derivatives that induce ferroptosis in breast cancer.

Compound	Strength of Evidence	Chemical subclass	Model/Disease	Effect (Concise Academic Wording)	Reference
Dihydroartemisinin (DHA)	Strong	Sesquiterpene lactone	Breast cancer (radiation model)	Targets ferroptosis signaling via hsa_circ_0001610 to increase radiosensitivity.	[144]
Dihydroisotanshinone I	Strong	Abietane diterpenoid (tanshinone family)	Breast cancer cells	Induces both ferroptosis and apoptosis.	[158]
Tanshinone IIA	Strong	Abietane diterpenoid	Breast cancer cells	Destabilizes SLC7A11 through KDM1A–PIAS4–mediated SUMOylation, promoting ferroptosis.	[146]
Curcumol derivative HCL-23	Strong	Sesquiterpenoid	TNBC	Induces HO-1-dependent ferroptosis and apoptosis, inhibiting malignant phenotype.	[135]
Curcumenol	Strong	Sesquiterpenoid	TNBC	Promotes ferroptosis via the SLC7A11/NF-κB/TGF-β pathway and suppresses malignant progression.	[147]
Anomanolide C	Strong	Sesquiterpene lactone	TNBC	Triggers autophagy-dependent ferroptosis by promoting GPX4 ubiquitination, suppressing progression and metastasis.	[148]
Taraxerol	Strong	Triterpenoid	Breast cancer cells	Inhibits Nrf2 transcriptional activity, facilitating MIB2-mediated GPX4 ubiquitination and ferroptosis.	[149]
Eupaformosanin	Strong	Sesquiterpene lactone	TNBC	Induces apoptosis and ferroptosis through ubiquitination of mutant p53.	[150]
Ursolic acid	Strong	Triterpenoid	TNBC stem-like cells	Inhibits proliferation through NRF2-mediated ferroptosis.	[151]
Hinokitiol	Strong	Monoterpenoid (tropolone) complex	TNBC (in vitro and in vivo)	Acts as a ferroptosis inducer to inhibit tumor growth.	[152]
Oridonin	Strong	Diterpenoid	Breast cancer cells	Potentiates RSL3-induced ferroptosis via JNK/Nrf2/HO-1 oxidative-stress signaling.	[153]

### 4.3. Saponins

This chapter examines the potential of *Saponins* as ferroptosis inducers. The key experimental evidence and mechanisms of action reported in the literature are compiled in Table 4. Complementing this information, Figure 5 provides a visual reference for the chemical structures of notable compounds from this class.

Saponins, distinguished by their amphiphilic architectures and varied bioactivities, have surfaced as possible inducers of ferroptosis in breast cancer research [159]. In contrast to chemicals that interact with a singular molecular target, saponins frequently influence several signaling pathways or targets the regulation of ferroptosis. Certain studies indicate a correlation between ferroptosis and other mechanisms of cell death or the regulation of the tumor microenvironment, implying that saponins could serve as multifunctional anticancer lead compounds [160].

Albiziabioside A exemplifies the multifunctionality of saponins. A chemically optimized derivative developed by Huang et al. functions as a p53 activator while concurrently triggering caspase-dependent apoptosis and ferroptosis in breast cancer cells [161]. In another study, Zhang et al. conjugated Albiziabioside A with the pyruvate-dehydrogenase-kinase inhibitor dichloroacetate in a separate study, establishing a synergistic treatment approach that induces apoptosis, initiates ferroptosis, and encourages M2-type tumor-associated macrophage (TAM) polarization, thereby producing anticancer effects through direct cytotoxicity and immune-microenvironment remodeling [162].

While comprehensive data on the ginsenoside family and ferroptosis in breast cancer are scarce, research on particular fractions or individual saponins has shown potential mechanisms. Kim et al. demonstrated that red-ginseng polysaccharides extracted from Panax ginseng down-regulate GPX4 and cause ferroptosis in breast cancer cells [163]. Similarly, Wang et al. observed that notoginsenoside R1 (NGR1) from Panax notoginseng inhibitsRUNX2 and modulates the AGE–RAGE pathway, thereby facilitating ferroptosis [164].

Multi-pathway control by saponins has been validated in many compounds that influence fundamental ferroptosis pathways. Yan and Xuan indicated that Paris saponin VII suppresses the NRF2/GPX4 pathway [165], while Chen et al. demonstrated that formosanin C increases the overall vulnerability of TNBC cells [166]. Other saponins affect new regulators: Wang et al. found that kinsenoside induces ferroptosis in TNBC via inhibiting diacylglycerol acyltransferase 1 (DGAT1)-mediated lipid-droplet formation [167].

Two primary characteristics encapsulate the function of saponins in promoting ferroptosis in breast cancer. Their synergistic ability to augment other therapy is paramount. Their capacity to function through various mechanisms or in conjunction with other cell-death pathways. For instance, Li et al. demonstrated that escin causes ferroptosis while synergizing with cisplatin to produce a heightened anticancer effect [168]. By combining direct cytotoxicity, chemo sensitization, and modification of the tumor microenvironment, emerge as attractive candidates for breast cancer combination therapy regimens [159,160].

**Table 4 biomolecules-15-01308-t004:** Saponins that induce ferroptosis in breast cancer.

Compound	Strength of Evidence	Chemical Class	Model/ Disease	Effect	Reference
Formosanin C	Strong	Steroidal saponin	MDA-MB-231 (TNBC), MCF-7 (Luminal A)	Induces ferroptosis characterized by elevated ROS and MDA, depletion of glutathione, and increased intracellular ferrous iron.	[166]
Paris saponin VII	Strong	Steroidal saponin	MCF-7 (ER^+^)	Promotes ferroptosis by inhibiting the Nrf2/GPX4 axis and intensifying oxidative lipid damage.	[165]
Kinsenoside	Strong	Diterpenoid saponin	MDA-MB-231 (TNBC)	Triggers ferroptosis by inhibiting DGAT1-mediated lipid droplet biogenesis and enhancing lipid peroxidation.	[167]
Escin	Strong	Steroidal saponin	Breast cancer cell lines	Elicits ferroptosis through lipid peroxidation and synergizes with cisplatin to enhance antitumor efficacy.	[168]
Notoginsenoside R1 (NGR1)	Strong	Steroidal saponin	Breast cancer cell lines	Accelerates ferroptosis by repressing RUNX2 and modulating the AGE-RAGE pathway, thereby amplifying oxidative stress.	[164]

### 4.4. Alkaloids

This section highlights key members of the *Alkaloids* family that have been shown to induce ferroptosis. The specific activities and mechanistic insights for each compound are cataloged in Table 5, while their chemical structures is presented in Figure 6.

Alkaloids are nitrogen-containing heterocyclic natural compounds characterized by varied chemical structures and broad biological activity, rendering them crucial frameworks in medication development [169]. Evidence suggests that certain alkaloids and their derivatives may function as ferroptosis inducers in breast cancer.

Recent studies have concentrated on employing alkaloids as organic ligands to develop medicinal metal complexes. Wang et al. developed a cycloplatinated compound (Pt-3) derived from an isoquinoline alkaloid, demonstrating superior anticancer efficacy compared to cisplatin in TNBC cells; the mechanism entails the activation of ferritinophagy and the induction of ferritinophagy-dependent ferroptosis [170]. Similarly, Lu et al. prepared an iridium(III) complex derived from an isoquinoline alkaloid that induces cell death through autophagy-dependent ferroptosis while simultaneously triggering immunogenic cell death (ICD) and inhibiting IDO, thereby offering experimental support for strategies that integrate ferroptosis induction with immune modulation in TNBC [171].

Besides synthetic derivatives, certain unaltered natural alkaloids directly trigger ferroptosis. Yi et al. indicated that peiminine—an active constituent of Fritillaria—induces ferroptosis via the NRF2 pathway and inhibits the growth of breast cancer cells [172]. NRF2 is typically regarded as a ferroptosis suppressor; this observation implies that NRF2 signaling may assume intricate or non-canonical roles under specific chemical manipulations. Wang et al. showed that the indole alkaloid indirubin successfully induces ferroptosis and inhibits tumor growth in vitro and in vivo in a 4T1 mouse breast cancer model [173].

These investigations elucidate two primary mechanisms by which alkaloids influence ferroptosis in breast cancer. Initially, alkaloid-based derivatives, functioning as ligands in metal complexes, can induce ferroptosis while concurrently engaging additional processes, including immunological activation. Secondly, natural alkaloids can directly induce ferroptosis by modulating additional critical signaling pathways, including NRF2. The combination of structural modifiability and intrinsic ferroptosis-inducing activity positions alkaloids render them intriguing lead molecules for developing ferroptosis-targeted therapies against breast cancer.

**Table 5 biomolecules-15-01308-t005:** Alkaloids that induce ferroptosis in breast cancer.

Compound	Strength of Evidence	Chemical Class	Model/ Disease	Effect	Reference
Peiminine	Strong	Natural alkaloid	Breast cancer cells	Induces ferroptosis through modulation of the Nrf2 pathway, leading to oxidative and lipid peroxidative stress.	[172]
Indirubin	Strong	Indole alkaloid	4T1 murine breast cancer (in vitro and in vivo)	Suppresses tumor growth by inducing ferroptosis marked by GPX4 downregulation and lipid peroxidation.	[173]

### 4.5. Other Chemical Classes

A variety of Other Chemical Classes have demonstrated the ability to trigger ferroptosis in breast cancer. Table 6 details these compounds and their underlying mechanisms, while Figure 7 displays the chemical structures of representative examples.

In addition to the aforementioned categories, polysaccharides, quinones, and sterols also play a role in modulating ferroptosis in breast cancer, providing diverse molecular entities for targeted pharmacological development.

Growing data indicates that, alongside their immunomodulatory effects, macromolecular polysaccharides can affect ferroptosis. Wang et al. initially showed that *Lycium barbarum* polysaccharide directly triggers ferroptosis in breast cancer cells, thereby introducing a novel mechanism to its anticancer activity [174]. Polysaccharides demonstrate potential as chemosensitizers: Yang et al. discovered that *Tetrastigma hemsleyanum* polysaccharide enhances the efficacy of doxorubicin to promote ferroptosis in TNBC cells while influencing host immunity, indicating a viable “ferroptosis–immunity” combination strategy [92].

Quinones, capable of inducing lipid peroxidation through redox cycling, are pivotal in ferroptosis regulation [175]. While research on breast cancer is scarce, pathways identified in other cancers provide valuable insights. Plumbagin and juglone promote ferroptosis by blocking GPX4 [176], while β-lapachone enhances the labile iron pool through NCOA4-mediated ferritinophagy to initiate ferroptosis in colorectal cancer [177]. In breast cancer specifically, de Carvalho et al. showed that bromo-naphthoquinone combined with tannic acid yields synergistic antitumor effects in TNBC cells [178]. Significantly, quinone activity depends largely on NAD(P)H:quinone oxidoreductase-1 (NQO1) expression. Wang et al. discovered that tanshinone can inhibit ferroptosis in NQO1-high cells [179], suggesting that NQO1 status may determine whether quinones function as pro-death or protective agents in breast cancer [180].

Sterols, especially cardiac glycosides, are transitioning from cardiovascular to anticancer research [181,182]. Recent research delineates their molecular targets in ferroptosis associated with breast cancer. Wei et al. established that bufalin induces ferroptosis via the 2,4-dienoyl-CoA reductase 1(DECR1)–SLC7A11 axis [35]. Given that DECR1 has been identified as a key survival factor protecting prostate tumor cells from ferroptosis, targeting it appears to be a promising therapeutic strategy [183]. Zhang et al. established DECR1 as the principal target of ergosterol B in ferroptosis induction [184], thereby affirming DECR1 as a common sterol target.

These unique natural-product frameworks stimulate innovative pharmaceutical innovations. Using bufalin as a chemical backbone, researchers created a photo-activated methylene-blue-bufalin conjugate. Wu et al. demonstrated that this compound selectively targets GPX4 for degradation and, under light exposure, effectively triggers ferroptosis-like death in breast-cancer cells [185]. Integrating natural-product scaffolds with techniques like photodynamic therapy is a pioneering approach for creating extremely specific ferroptosis inducers.

The ferroptosis findings for these quinone compounds mostly originate from non–breast cancer tumor models; nonetheless, their mechanisms of action demonstrate significant translational potential in breast cancer and may support the development of new targeted therapies.

**Table 6 biomolecules-15-01308-t006:** Other chemical classes that induce ferroptosis in breast cancer.

Compound	Strength of Evidence	Chemical Class	Model/ Disease	Effect	Reference
*Tetrastigma hemsleyanum* polysaccharide + doxorubicin	Strong	Polysaccharide (combination)	TNBC cells	Produces a synergistic antitumor effect by promoting ferroptosis and modulating antitumor immunity.	[92]
Plumbagin	Moderate	Quinone	(Glioma; mechanistic reference)	Acts as a GPX4-targeting ferroptosis inducer, offering mechanistic insight transferable to breast cancer contexts.	[176]
Juglone	Moderate	Quinone	(Endometrial cancer; mechanistic reference)	Induces ferroptosis via GPX4 inhibition, providing a mechanistic template relevant to breast cancer.	[186]
β-Lapachone	Moderate	Quinone	Colorectal cancer	Activates NCOA4-mediated ferritinophagy and JNK signaling to trigger ferroptosis; the mechanism is informative for breast cancer translation.	[177]
Bufalin	Strong	Sterol (cardiac glycoside)	Breast cancer cells	Induces ferroptosis by perturbing the DECR1–SLC7A11 axis and enhancing lipid peroxidation.	[35]
Erigoster B	Strong	Sterol	Breast cancer cells	Targets DECR1 to reprogram phosphatidylcholine/arachidonic acid metabolism and enforce ferroptosis.	[184]

Ferroptosis findings for these quinone compounds mostly originate from non-breast cancer tumor models; nonetheless, their mechanisms of action demonstrate significant translational potential in breast cancer and may support the development of novel targeted therapies.

## 5. Conclusion, Current Limitations, and Future Perspectives

### 5.1. A Critical Review of Gaps and Limitations

A core challenge of natural products lies in their inherent multi-target pharmacological properties. While this characteristic may offer potential advantages for synergistic therapy, it also introduces complexity to the traditional, single-target-based drug discovery paradigm [169,187]. For instance, the pro-ferroptotic effects of curcumin are linked to multiple signaling pathways and broad pro-oxidant activities, making the precise identification of a dominant mechanism highly challenging [188]. To address this, Network Pharmacology offers a powerful solution. By integrating data from medicinal chemistry, genomics, and bioinformatics, researchers can construct complex protein–protein interaction (PPI) networks and analyze them within the context of specific diseases like breast cancer [189]. Subsequently, topological analysis of this network can identify key hub genes with the highest connectivity in regulating ferroptosis [190]. This process transforms a once-vague multi-target problem into a testable scientific hypothesis focused on a few high-priority molecules [125], providing both a macroscopic and precise guide for subsequent mechanistic research and drug optimization [191].

Beyond mechanistic uncertainty, the clinical development of natural products is fundamentally challenged by their poor pharmacokinetic properties. Many lead compounds exhibit extremely low oral bioavailability, preventing them from reaching therapeutically relevant concentrations in the blood. For example, although curcumin displays potent anticancer activity in vitro, its clinical application is severely limited by a bioavailability often reported to be less than 1%, primarily due to extensive first-pass metabolism [192]. Resveratrol is another classic example of this process; its rapid glucuronidation and sulfation in the liver lead to similarly low bioavailability and a very short plasma half-life, making it impossible to maintain stable therapeutic concentrations via oral administration [193,194]. Furthermore, the use of non-standardized plant extracts introduces significant batch-to-batch variability, undermining research reproducibility and posing a major regulatory hurdle for clinical translation [195].

In comprehensively evaluating the current literature on natural product-induced ferroptosis, maintaining a critical perspective on the methodological rigor of preclinical studies is essential. Although existing research has provided valuable insights, some studies, particularly early exploratory work, may have limitations in data presentation and statistical analysis. For instance, many conclusions relying on Western blots sometimes lack sufficient quantification and an adequate number of biological replicates to support their reliability. Similarly, in some animal studies, small sample sizes (n-numbers) may limit their statistical power, making the generalizability of the conclusions questionable. Moreover, many findings are highly dependent on a few specific cancer cell lines, which may not fully reflect the high degree of heterogeneity in breast cancer among actual patients. Therefore, future research must, while pursuing mechanistic innovation, place greater emphasis on adhering to rigorous experimental standards to ensure the reliability and reproducibility of results, thereby laying a more solid foundation for future clinical translation.

It is important to note that, to date, no clinical trials have been specifically designed to evaluate natural products as ferroptosis modulators in breast cancer, and the current body of evidence remains exclusively at the preclinical stage. The reliance on oversimplified preclinical models is another major obstacle hindering translational success. The vast majority of studies are conducted on two-dimensional (2D) cell monolayers, a system that fails to replicate the complex three-dimensional architecture and cell–cell interactions of solid tumors, nor can it accurately assess drug resistance, phenomena better captured by 3D spheroids and organoid systems [196,197]. More importantly, these models completely ignore the influence of the tumor microenvironment (TME), which has been recognized as a key regulator of ferroptosis [198]. Studies have shown that stromal cells (such as cancer-associated fibroblasts) can actively confer ferroptosis resistance to neighboring cancer cells, a critical biological interaction completely absent in standard in vitro experiments [21]. Finally, these models fail to account for intra-tumoral heterogeneity. A multi-omics study has revealed distinct and heterogeneous ferroptosis phenotypes within triple-negative breast cancer, suggesting that a drug effective against one cell subpopulation may be ineffective against others within the same tumor [47]. This oversimplification likely leads to a systematic overestimation of candidate drug efficacy.

This translational gap is further exacerbated by the lack of validated biomarkers for patient stratification and treatment monitoring. Current development of predictive biomarkers has mainly produced computational gene signatures (such as FERscore) based on retrospective dataset analysis, but they lack the prospective clinical validation necessary to guide patient selection [199,200]. A second major gap is the absence of pharmacodynamic (PD) biomarkers; there are currently no established non-invasive methods (such as specific imaging agents) to confirm whether a therapeutic agent has engaged its target and initiated ferroptosis within a patient’s tumor [201]. Finally, the mechanisms regulating resistance are poorly understood, and there are no biomarkers to identify patients with primary resistance or to monitor the emergence of resistant subclones [71,202]. This series of deficiencies in predictive, pharmacodynamic, and resistance biomarkers makes the rational design of clinical trials impossible and constitutes a fundamental barrier to the progress of this therapeutic strategy.

However, the most fundamental limitation of targeting ferroptosis lies in the potential for on-target toxicity in non-cancer tissues. This risk stems from the fact that ferroptosis is a basic physiological process, not a tumor-specific pathway; therefore, the same mechanisms activated in cancer cells can also disrupt homeostasis in normal tissues. The heart is one of the most vulnerable high-risk organs, as its energy metabolism is highly dependent on mitochondrial iron–sulfur clusters, the destabilization of which directly leads to functional impairment. This vulnerability is exemplified by the clinical cardiotoxicity of doxorubicin: a study by Liao et al. showed that the drug triggers ferroptosis by promoting lipid peroxidation in cardiomyocytes [203], while Cantrell et al. further confirmed that the protective effects of ferroptosis inhibitors rely on the direct stabilization of these mitochondrial iron–sulfur clusters [204]. The kidneys also exhibit significant susceptibility, as their renal tubular epithelial cells are rich in polyunsaturated fatty acids (PUFAs), making them prime substrates for lipid peroxidation [205], and inhibiting the key enzyme ACSL4 has been shown to effectively mitigate ferroptotic damage in the kidneys [206]. Furthermore, the central nervous system is highly sensitive to ferroptosis imbalance due to its high iron content. Studies in neurodegenerative disease models have revealed that GPX4 dysfunction is a key determinant of neuronal ferroptosis [31], and pharmacological inhibition of this pathway can alleviate neurotoxicity [207].

### 5.2. Future Perspectives and Strategies to Surmount Obstacles

Medicinal chemistry optimization is foundational for converting promising natural products into viable clinical candidates. The strategy often begins with identifying natural product scaffolds that modulate ferroptosis and then systematically modifying them to develop potent and selective inhibitors, with a primary focus on the key regulator, glutathione peroxidase 4 (GPX4) [137,208]. Recent efforts have successfully yielded novel derivatives with enhanced anticancer activity. For example, selenium-containing derivatives of artesunate have been synthesized to effectively induce GPX4-mediated ferroptosis [209]. Furthermore, diversity-oriented synthesis has been employed to discover novel covalent GPX4 inhibitors and ferrocenophane-appended derivatives, which demonstrate high selectivity and improved drug-like properties [210,211]. These synthetic efforts are increasingly guided by computational methods, such as molecular docking, which help predict interactions and identify potential allosteric inhibition sites on targets like GPX4 [212].

Innovative drug delivery technologies are crucial for translating the potential of natural ferroptosis inducers into clinical reality, with developments extending far beyond traditional liposomes and polymer micelles. Recent advances have focused on sophisticated biomimetic nanoplatforms, which are camouflaged with membranes derived from macrophages or cancer cells to evade immune clearance and achieve superior tumor-homing capabilities [213,214,215]. Beyond passive targeting, these nanotechnologies are also engineered to actively remodel the tumor microenvironment (TME) to amplify the ferroptotic effect. This can be achieved through various strategies, such as depleting intracellular glutathione (GSH) to dismantle the cell’s primary antioxidant defense system [216,217], or delivering iron ions to catalyze the Fenton reaction, ensuring a continuous supply of cytotoxic reactive oxygen species [218]. Furthermore, smart designs like size-switchable nanocapsules are being developed to overcome physical barriers and enhance deep tumor penetration [219,220].

Combination tactics based on mechanisms are crucial for overcoming the drug resistance that frequently limits the efficacy of single-agent therapies [137]. A particularly promising strategy is to leverage ferroptosis-inducing natural products to re-sensitize drug-resistant breast cancer cells to standard chemotherapies. For instance, studies have shown that traditional herbal formulas like Danggui Buxue Tang can synergize with doxorubicin by inducing ferroptosis in triple-negative breast cancer (TNBC) [221]. Similarly, novel taxane derivatives have been demonstrated to overcome paclitaxel resistance by triggering a noncanonical ferroptosis pathway [90]. The core principle of this approach is that activating ferroptosis can serve as an alternative cell death mechanism to bypass apoptosis-resistance pathways, thereby enhancing chemosensitivity [222]. Clarifying the molecular foundations of this synergy is essential for designing rational combination regimens with chemotherapeutics, targeted agents, and immunotherapies.

Finally, future research must shift its focus from in vitro cell experiments to more clinically relevant models to bridge the current translational gap. A priority research direction is the validation of promising natural products in appropriate preclinical animal models. For example, Tanshinone IIA, which has shown potent ferroptosis-inducing activity in triple-negative breast cancer (TNBC) cells, could be administered via tail vein injection in mice bearing 4T1 orthotopic tumors [15]. Such a study should not only evaluate its efficacy in inhibiting primary tumor growth and lung metastasis but also confirm its in vivo ferroptosis-inducing mechanism by detecting levels of 4-HNE (a marker of lipid peroxidation) in tumor tissues [223]. Concurrently, conducting translational research based on human clinical samples is also crucial. A specific and feasible experiment would be to use tumor tissue microarrays (TMAs) containing samples from hundreds of breast cancer patients to quantitatively assess the expression level of the key negative ferroptosis regulator, GPX4, using immunohistochemistry (IHC) [102]. Subsequently, Kaplan–Meier analysis could be used to test whether low GPX4 expression is significantly associated with longer disease-free survival (DFS) in patients, particularly those receiving anthracycline-based chemotherapy [54,224].

## Figures and Tables

**Figure 1 biomolecules-15-01308-f001:**
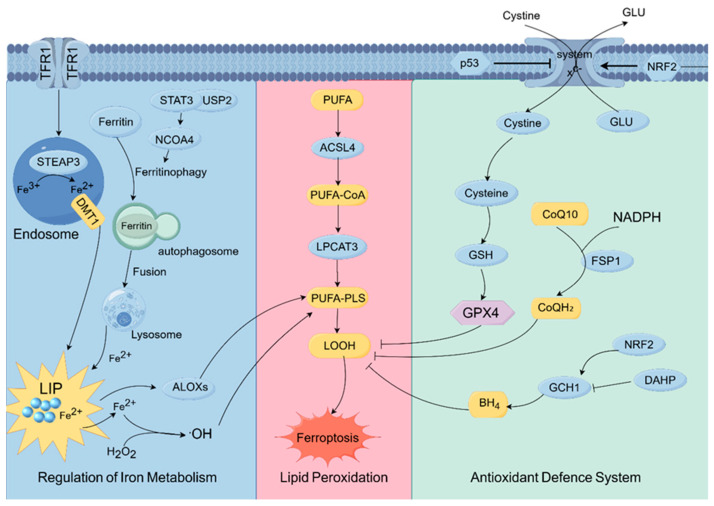
Molecular regulatory network of ferroptosis. This figure illustrates the three core pillars of ferroptosis regulation: the pro-ferroptotic processes of iron metabolism and lipid peroxidation, which are counteracted by a multi-layered antioxidant defense system centered on the GPX4, FSP1, and GCH1 pathways (by figdraw.com, accessed on 6 August 2025).

**Figure 2 biomolecules-15-01308-f002:**
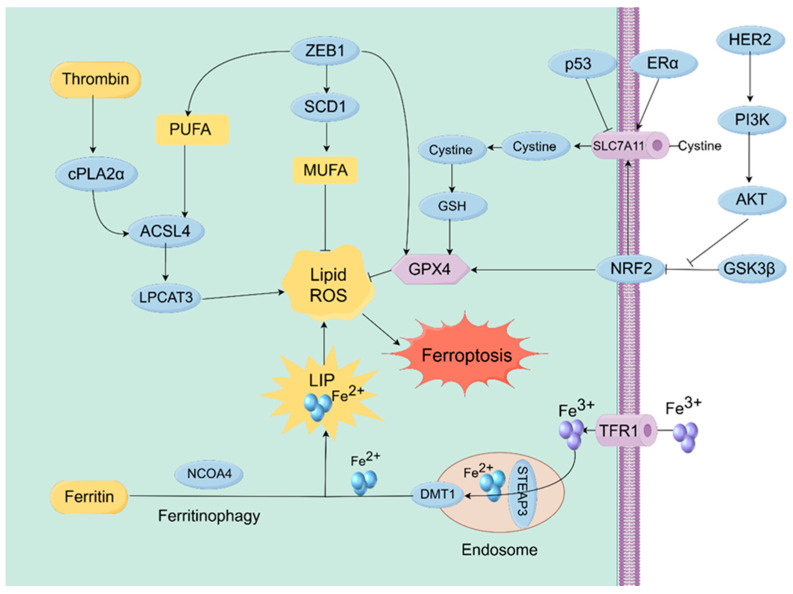
Molecular mechanisms regulating ferroptosis in breast cancer. Figure depicts how ferroptosis is regulated in breast cancer. Key subtype-specific pathways, including those driven by ZEB1, ERα, and HER2, directly control core ferroptotic processes to create unique therapeutic vulnerabilities (by figdraw.com, accessed on 6 August 2025).

**Figure 3 biomolecules-15-01308-f003:**
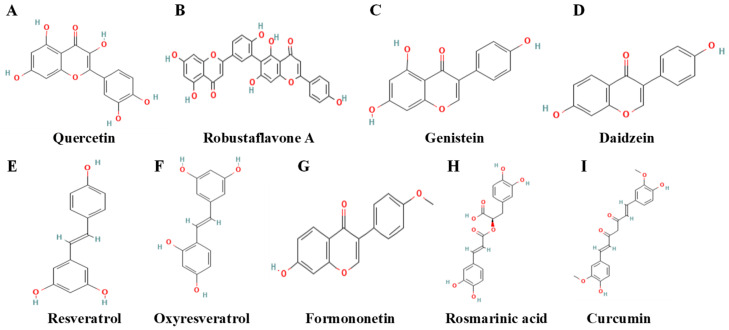
Chemical structures of representative polyphenolic compounds that modulate ferroptosis. All chemical structures were obtained from the PubChem database with the following Compound IDs (CIDs): (**A**) quercetin (5280343), (**B**) Robustaflavone A (14778159), (**C**) genistein (5280961), (**D**) daidzein (5281708), (**E**) resveratrol (445154), (**F**) oxyresveratrol (5281717), (**G**) Formononetin (5280378), (**H**) rosmarinic acid (5281792), and (**I**) curcumin (969616).

**Figure 4 biomolecules-15-01308-f004:**
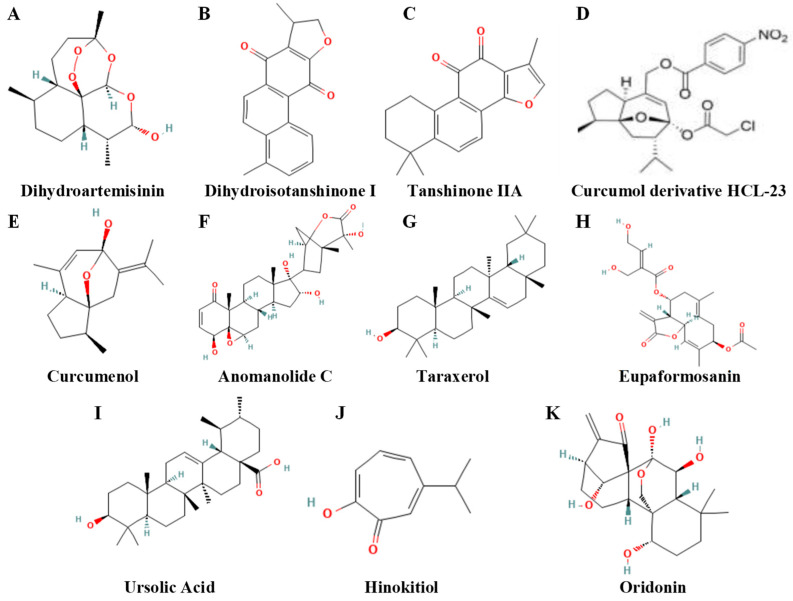
Chemical structures of representative terpenoids and their derivatives that modulate ferroptosis. Figure displays structures of: (**A**) dihydroartemisin (DHA), (**B**) dihydroisotanshinone I, (**C**) tanshinone IIA, (**D**) curcumol derivative HCL-23, (**E**) curcumenol, (**F**) anomanolide C, (**G**) taraxerol, (**H**) eupaformosanin, (**I**) ursolic acid, (**J**) hinokitiol, and (**K**) oridonin. Structure for (**D**) is shown as reported in [135]. All other chemical structures were obtained from the PubChem database. CIDs are as follows: 3000518 (dihydroartemisinin), 89406 (dihydroisotanshinone I), 164676 (tanshinone IIA), 167812 (curcumenol), 44423050 (anomanolide C), 92097 (taraxerol), 5281453 (eupaformosanin), 64945 (ursolic acid), 3611 (hinokitiol), and 5321010 (oridonin).

**Figure 5 biomolecules-15-01308-f005:**
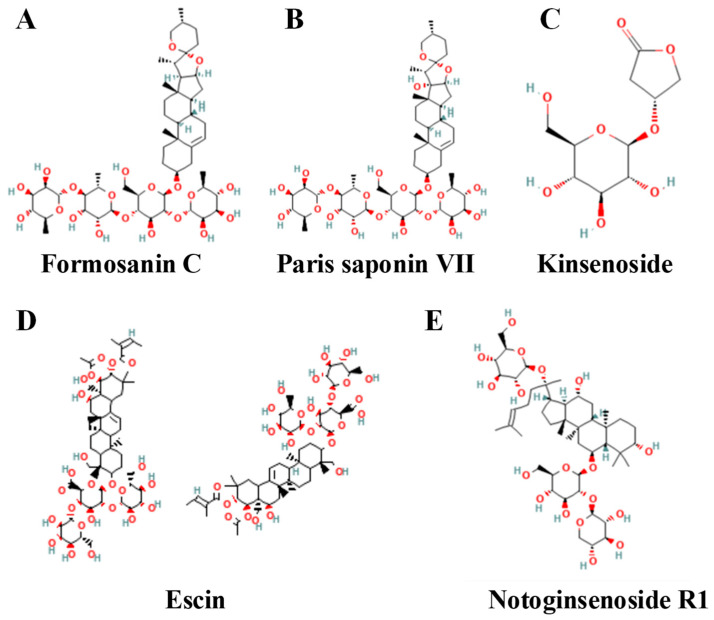
Chemical structures of representative saponins that modulate ferroptosis. All chemical structures were obtained from the PubChem database with the following Compound IDs (CIDs): (**A**) formosanin C (21603986), (**B**) Paris saponin VII (176233), (**C**) kinsenoside (10422896), (**D**) escin (6433489), and (**E**) notoginsenoside R1 (441934).

**Figure 6 biomolecules-15-01308-f006:**
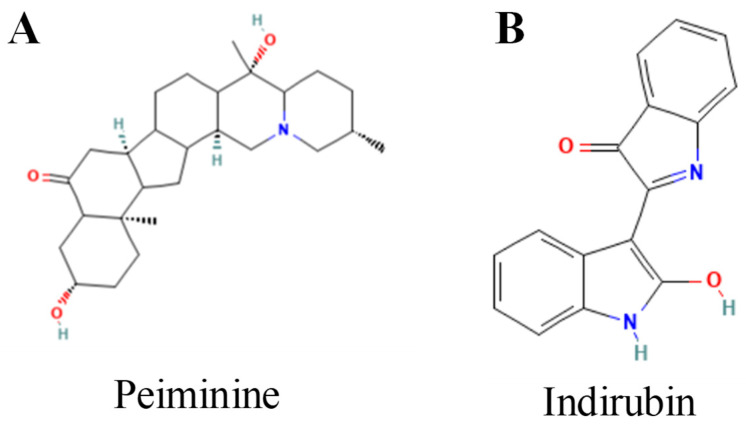
Chemical structures of representative alkaloids that modulate ferroptosis. All chemical structures were obtained from the PubChem database with the following Compound IDs (CIDs): (**A**) peiminine (5320446) and (**B**) indirubin (3707).

**Figure 7 biomolecules-15-01308-f007:**
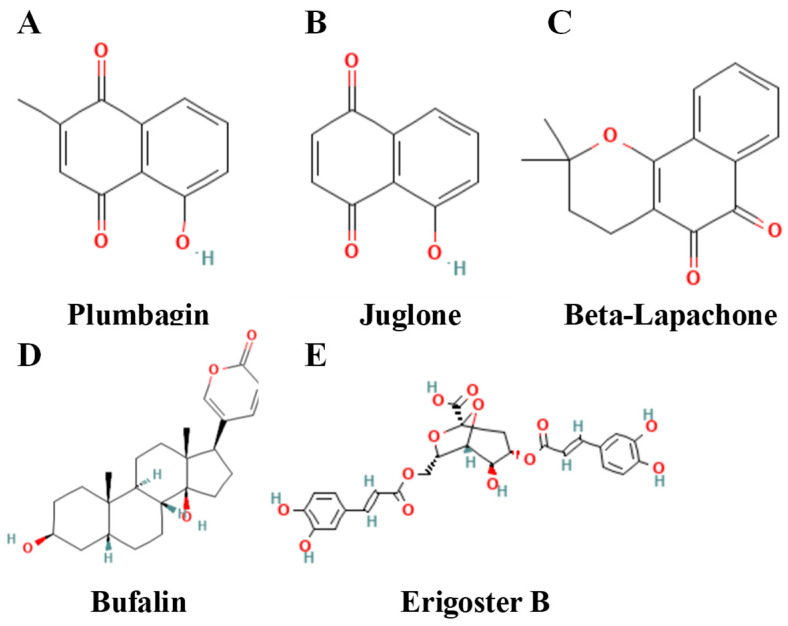
Chemical structures of representative other chemical classes of compounds that modulate ferroptosis. All chemical structures were obtained from the PubChem database with the following Compound IDs (CIDs): (**A**) plumbagin (10205), (**B**) juglone (3806), (**C**) β-lapachone (3885), (**D**) bufalin (9547215), and (**E**) erigoster B (137796506).

**Table 1 biomolecules-15-01308-t001:** Differential ferroptosis sensitivity and underlying mechanisms across breast cancer molecular subtypes.

Molecular Subtype	Baseline Ferroptosis Sensitivity	Key Mechanisms	Reference
TNBC	High	ZEB1 reshapes lipid composition (↓MUFA, ↑PUFA); mitomiR-3–ZEB1–GPX4 inhibition; salidroside amplifies lipid peroxidation via SCD1 inhibition and NCOA4-driven ferritinophagy	[51,64,66]
Luminal A/B (ER^+^/PR^+^)	Low/tolerant	ERα upregulates System Xc^−^ (SLC7A11, SLC3A2) to suppress ferroptosis; CDK4/6 inhibitor response depends on SLC7A11 downregulation; USP35–BRD4–SLC7A11 axis maintains antioxidant defense	[67,75,76]
HER2^+^	Medium–low	HER2–PI3K–AKT–mTOR enhances antioxidant capacity; integrin αvβ3 mediates dual resistance to TKIs and ferroptosis; irreversible HER2 inhibitors (e.g., neratinib) restore sensitivity to RSL3	[70,71]

Abbreviations: ↑, promotion; ↓, inhibition.

## Data Availability

Data sharing is not applicable to this article as no new data were created or analyzed in this study. All information discussed is sourced from the publications cited in the reference list.

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
