# Peer review of "Natural Compounds as Modulators of Ferroptosis: Mechanistic Insights and Therapeutic Prospects in Breast Cancer"

_biomolecules, 2025, doi:10.3390/biom15091308_

Round 1
Reviewer 1 Report
Comments and Suggestions for Authors
Minor comment#1:
While the manuscript provides a comprehensive overview of ferroptosis regulators in breast cancer, there appears to be a conceptual gap between Figure 1 and Figure 2. In Figure 1, lipid hydroperoxides (LOOH) are depicted as the central executors of ferroptotic cell death, reflecting the canonical mechanism. However, in Figure 2, LOOH seems largely absent or less emphasized in the context of breast cancer subtypes, which may confuse readers. It would be helpful to clarify that, in certain breast cancer subtypes—such as luminal or HER2-positive tumors—the accumulation of LOOH is more difficult to detect due to enhanced antioxidant defenses (e.g., elevated GPX4, SLC7A11 expression) and metabolic adaptations. Adding a brief discussion on the subtype-specific dynamics of LOOH formation and its detection could strengthen Section 3, highlighting why ferroptosis sensitivity varies and how LOOH serves as the ultimate biochemical effector, even if it is less apparent experimentally in breast cancer models.
Minor comment #2:
Chapter 4 lists natural products reported to induce ferroptosis in breast cancer. However, the rationale for focusing specifically on natural compounds, and the interest in their effects on iron metabolism, should be explicitly stated at the beginning of Chapter 4 or at the end of Chapter 3. This would help readers understand the purpose and context of the chapter.
Minor comment#3:
In Section 5, it would be helpful to include more concrete suggestions for future experiments. For instance, measuring iron accumulation and ferroptosis markers in human breast cancer biopsy specimens, or testing direct administration of natural products in appropriate mouse models. Highlighting what could be done immediately and which experimental approaches should be prioritized would strengthen the practical relevance of this section.
Author Response
|
Comments 1: While the manuscript provides a comprehensive overview of ferroptosis regulators in breast cancer, there appears to be a conceptual gap between Figure 1 and Figure 2. In Figure 1, lipid hydroperoxides (LOOH) are depicted as the central executors of ferroptotic cell death, reflecting the canonical mechanism. However, in Figure 2, LOOH seems largely absent or less emphasized in the context of breast cancer subtypes, which may confuse readers. It would be helpful to clarify that, in certain breast cancer subtypes—such as luminal or HER2-positive tumors—the accumulation of LOOH is more difficult to detect due to enhanced antioxidant defenses (e.g., elevated GPX4, SLC7A11 expression) and metabolic adaptations. Adding a brief discussion on the subtype-specific dynamics of LOOH formation and its detection could strengthen Section 3, highlighting why ferroptosis sensitivity varies and how LOOH serves as the ultimate biochemical effector, even if it is less apparent experimentally in breast cancer models. |
|
Response 1: We sincerely thank the reviewer for this excellent and insightful comment. We agree that clarifying the apparent conceptual gap regarding LOOH between the canonical pathway (Figure 1) and its role in specific breast cancer subtypes (Figure 2) is crucial. To address this thoroughly, we have added a detailed, evidence-based paragraph at the end of Section 3.2. This new text not only explains why LOOH is difficult to detect in resistant subtypes but also provides specific experimental evidence (e.g., the effects of GPX4 and SLC7A11 inhibitors like RSL3 and Erastin) and supporting references to substantiate our explanation of the dynamic equilibrium of LOOH. This strengthens the argument considerably. (3.2 Differences in Ferroptosis Sensitivity Across Molecular Subtypes of Breast Cancer, Lines 229-243) “Although lipid hydroperoxides (LOOH) are the ultimate biochemical executioners of ferroptosis, their direct detection is highly challenging in breast cancer subtypes with intrinsic resistance to this cell death pathway [73]. This resistance primarily stems from an efficient antioxidant system that relies on glutathione (GSH) biosynthesis and its function as a cofactor for the enzymatic activity of Glutathione Peroxidase 4 (GPX4) [55]. The functional potency of this system has been well-validated experimentally. For in-stance, direct pharmacological inhibition of GPX4's enzymatic activity with agents like RSL3, or depletion of GSH by blocking the upstream transporter SLC7A11 with inhibitors such as Erastin, both lead to a sharp increase in intracellular LOOH levels in breast cancer cells [74, 75]. This evidence strongly suggests that in many tumors with high expression of GPX4 and SLC7A11, such as certain luminal and HER2-positive subtypes, a dynamic equilibrium exists between LOOH production and clearance, keeping its steady-state concentration suppressed below detectable thresholds. Consequently, in these subtypes, the research focus naturally shifts from LOOH itself to the key upstream proteins that regulate its levels.” Refences: 73. Dixon, Scott J, and James A %J Nature reviews Molecular cell biology Olzmann. "The Cell Biology of Ferroptosis." 25, no. 6 (2024): 424-42. 74. Dixon, Scott J, Kathryn M Lemberg, Michael R Lamprecht, Rachid Skouta, Eleina M Zaitsev, Caroline E Gleason, Darpan N Patel, Andras J Bauer, Alexandra M Cantley, and Wan Seok %J cell Yang. "Ferroptosis: An Iron-Dependent Form of Nonapoptotic Cell Death." 149, no. 5 (2012): 1060-72. 75. Yang, Wan Seok, Rohitha SriRamaratnam, Matthew E Welsch, Kenichi Shimada, Rachid Skouta, Vasanthi S Viswanathan, Jaime H Cheah, Paul A Clemons, Alykhan F Shamji, and Clary B %J Cell Clish. "Regulation of Ferroptotic Cancer Cell Death by Gpx4." 156, no. 1 (2014): 317-31."
|
|
Comments 2: Chapter 4 lists natural products reported to induce ferroptosis in breast cancer. However, the rationale for focusing specifically on natural compounds, and the interest in their effects on iron metabolism, should be explicitly stated at the beginning of Chapter 4 or at the end of Chapter 3. This would help readers understand the purpose and context of the chapter. |
|
Response 2: We thank the reviewer for this valuable suggestion. We agree that providing a clear rationale is essential. Accordingly, we have added a new introductory paragraph at the beginning of Section 4 to explicitly state the purpose and context for focusing on natural products. Details are listed as follows. (4.Natural Products as Modulators of Ferroptosis in Breast Cancer,Lines 324-343) “As a distinct form of non-apoptotic regulated cell death, ferroptosis offers a novel strategy to overcome key challenges in breast cancer, as its molecular mechanism can effectively bypass obstacles such as therapeutic resistance and immune evasion. The complex molecular network governing ferroptosis, in turn, provides a wealth of molecular targets for precision therapeutic intervention. Among various strategies, natural products have garnered significant attention as a class of highly promising modulators. Their value lies primarily in their role as a source of established drug lead compounds, often possessing favorable safety profiles and inherent multi-target regulatory capabilities. Therefore, this chapter will systematically review the natural compounds reported to date that can induce or sensitize breast cancer cells to ferroptosis. However, before delving into the specific biological activities of these various natural compounds, it is crucial to first acknowledge a prevalent and critical challenge: their common pharmacokinetic deficiencies. Many promising natural products, including highly-studied molecules like curcumin and resveratrol, are hampered by issues such as low aqueous solubility, poor intestinal absorption, and rapid in vivo metabolism. These factors collectively lead to low bioavailability, making it extremely difficult to achieve and maintain effective therapeutic concentrations at the tumor site. Therefore, understanding this fundamental bottleneck is essential for objectively evaluating the preclinical data discussed hereafter and for recognizing the necessity of developing optimization strategies, such as advanced delivery systems.”
Comments 3: In Section 5, it would be helpful to include more concrete suggestions for future experiments. For instance, measuring iron accumulation and ferroptosis markers in human breast cancer biopsy specimens, or testing direct administration of natural products in appropriate mouse models. Highlighting what could be done immediately and which experimental approaches should be prioritized would strengthen the practical relevance of this section. Response 3: We thank the reviewer for this suggestion to improve the practical relevance of our manuscript. We agree that the "Future Perspectives" section would benefit from more concrete and prioritized experimental directions. Accordingly, we have substantially revised this section. This revision incorporates the reviewer's valuable examples, such as validating promising natural products in appropriate mouse models (e.g., Tanshinone IIA in a 4T1 orthotopic model) and conducting translational studies on human clinical samples (e.g., IHC analysis of GPX4 on TMAs). We have also emphasized that these foundational studies should be prioritized. Details are listed as follows (Conclusion, Current Limitations, and Future Perspectives, Lines 765-780): “Finally, future research must shift its focus from in vitro cell experiments to more clinically relevant models to bridge the current translational gap. A priority research direction is the validation of promising natural products in appropriate preclinical animal models. For example, Tanshinone IIA, which has shown potent ferroptosis-inducing activity in triple-negative breast cancer (TNBC) cells, could be administered via tail vein injection in mice bearing 4T1 orthotopic tumors [15]. Such a study should not only evaluate its efficacy in inhibiting primary tumor growth and lung metastasis but also confirm its in vivo ferroptosis-inducing mechanism by detecting levels of 4-HNE (a marker of lipid peroxidation) in tumor tissues [228]. Concurrently, con-ducting translational research based on human clinical samples is also crucial. A specific and feasible experiment would be to use tumor tissue microarrays (TMAs) containing samples from hundreds of breast cancer patients to quantitatively assess the expression level of the key negative ferroptosis regulator, GPX4, using immunohisto-chemistry (IHC)[104]. Subsequently, Kaplan-Meier analysis could be used to test whether low GPX4 expression is significantly associated with longer disease-free sur-vival (DFS) in patients, particularly those receiving anthracycline-based chemotherapy [55, 229].” Refences: “15. Ge, Anqi, Qi He, Da Zhao, Yuwei Li, Junpeng Chen, Ying Deng, Wang Xiang, Hongqiao Fan, Shiting Wu, Yan %J Journal of Cellular Li, and Molecular Medicine. "Mechanism of Ferroptosis in Breast Cancer and Research Progress of Natural Compounds Regulating Ferroptosis." 28, no. 1 (2024): e18044. 55. Sha, Rui, Yaqian Xu, Chenwei Yuan, Xiaonan Sheng, Ziping Wu, Jing Peng, Yaohui Wang, Yanping Lin, Liheng Zhou, and Shuguang %J EBioMedicine Xu. "Predictive and Prognostic Impact of Ferroptosis-Related Genes Acsl4 and Gpx4 on Breast Cancer Treated with Neoadjuvant Chemotherapy." 71 (2021). 104. Gong, Rong, Xiaoya Wan, Shilong Jiang, Yidi Guan, Yizhi Li, Ting Jiang, Zonglin Chen, Changxin Zhong, Linhao He, Zhongyuan %J Cell Death Xiang, and Differentiation. "Gpx4-Autac Induces Ferroptosis in Breast Cancer by Promoting the Selective Autophagic Degradation of Gpx4 Mediated by Traf6-P62." (2025): 1-16. 228. Zhong, Tianfei, Ying Li, Meng Jin, Jingqun Liu, Zhenyu Wu, Feiye Zhu, Lisha Zhao, Yongsheng Fan, Li Xu, Jinjun %J Cell Death Ji, and Disease. "Downregulation of 4-Hne and Foxo4 Collaboratively Promotes Nsclc Cell Migration and Tumor Growth." 15, no. 7 (2024): 546. 229. Zhao, Jiazheng, Ning Zhang, Xiaowei Ma, Ming Li, and Helin %J Cell Death Discovery Feng. "The Dual Role of Ferroptosis in Anthracycline-Based Chemotherapy Includes Reducing Resistance and Increasing Toxicity." 9, no. 1 (2023): 184.”
|
Reviewer 2 Report
Comments and Suggestions for Authors
Please, see the attached document.

Author Response
|
Response 1: Thank you for pointing this out. We wholeheartedly agree that explicitly distinguishing the origin of experimental evidence is crucial for scientific accuracy. To address your concern, we have now thoroughly revised the manuscript to clarify whether mechanistic claims are supported by direct breast cancer data or are extrapolated from other contexts. Firstly, the majority of the mechanisms, especially in chapters three and four, are derived from breast cancer. For example, in chapter four, all natural compounds except for Quinones have evidence originating from breast cancer. Secondly, we will connect this with the following.To achieve this, we first addressed foundational mechanistic principles that are observed across multiple cancer types. Instead of presenting them as breast-cancer-specific findings, we have added explicit phrasing to frame them as general oncological principles. This approach clarifies their broad applicability while still connecting them to the context of breast cancer. Details are listed as follows. (3. The Intrinsic Link Between Ferroptosis and Breast Cancer Therapy, Lines 163-164) "As a general principle in oncology, metabolic remodeling in cancer cells heightens reliance on iron and lipids, hence creating a foundational susceptibility to ferroptosis [46, 47]." Furthermore, we addressed the second scenario, where specific mechanistic claims are primarily supported by experimental data from non-breast cancer tumor types. In these cases, we have meticulously revised the text to explicitly name the original tumor type of the study. This ensures complete transparency and prevents any misinterpretation about the source of the evidence. Details are listed as follows. (4.5 Other Chemical Classes, Lines 616-618) "Given that DECR1 has been identified as a key survival factor protecting prostate tumor cells from ferroptosis, targeting it appears to be a promising therapeutic strategy [187]." References: “46. Peng, Cuixin, Yanning Chen, and Mingzhang %J Frontiers in Oncology Jiang. "Targeting Ferroptosis: A Promising Strategy to Overcome Drug Resistance in Breast Cancer." 14 (2024): 1499125. 47. Ye, Lvlan, Xiangqiong Wen, Jiale Qin, Xiang Zhang, Youpeng Wang, Ziyang Wang, Ti Zhou, Yuqin Di, Weiling %J Cell Death He, and Disease. "Metabolism-Regulated Ferroptosis in Cancer Progression and Therapy." 15, no. 3 (2024): 196. 187. Nassar, Zeyad D, Chui Yan Mah, Jonas Dehairs, Ingrid JG Burvenich, Swati Irani, Margaret M Centenera, Madison Helm, Raj K Shrestha, Max Moldovan, and Anthony S %J Elife Don. "Human Decr1 Is an Androgen-Repressed Survival Factor That Regulates Pufa Oxidation to Protect Prostate Tumor Cells from Ferroptosis." 9 (2020): e54166.”
|
|
Comments 2: I suggest you improve grammar consistency, e.g., “cancer is the predominant tumor” (Abstract, line 11) should be phrased more scientifically as “Breast cancer is the most prevalent malignant tumor in women.” |
|
Response 2: We thank the reviewer for this valuable suggestion. We agree that the original phrasing was not sufficiently scientific. Accordingly, we have revised this sentence (Abstract, line 12) to “Breast cancer is the most prevalent malignant tumor in women.” We have also carefully proofread the entire manuscript to correct any other similar issues and improve grammatical consistency.
Comments 3: Please, can you make the abstract more precise? It currently repeats ideas about resistance and metastasis without clear novelty. Response 3: Thank you for your valuable feedback. We agree that the original abstract was repetitive in its discussion of resistance and metastasis, which obscured the novelty of our review. To address this, we have substantially revised the abstract. We consolidated the description of the clinical challenges (resistance, metastasis, and immune evasion) into a single, more concise and impactful statement. This revision allows us to introduce ferroptosis more directly as a novel strategy that simultaneously addresses these core challenges, thereby better highlighting the central value and novelty of our review. Details are listed as follows. (Abstract, Lines 12-27) “Breast cancer is the most prevalent malignant tumor in women. However, its clinical management is severely hindered by three interconnected challenges that limit long-term survival: treatment resistance, metastatic dissemination, and immunological evasion. Ferroptosis, an iron-dependent form of regulated cell death, is emerging as a novel strategy to overcome these obstacles. Furthermore, it demonstrates significant potential in inhibiting tumor metastasis and modifying the anti-tumor immune microenvironment, which provides a novel approach to address the core dilemma of breast cancer. Natural products have emerged as significant sources of ferroptosis inducers owing to their distinctive chemical variety, multi-target regulatory capabilities, and acceptable safety profile. Increasing data indicates that several natural compounds can function as effective inducers or sensi-tizers of ferroptosis cell death. This review provides a thorough evaluation of current progress in harnessing natural ingredients to trigger ferroptosis for breast-cancer treatment. It also elucidates the fundamental molecular mechanisms. Furthermore, it encapsulates therapeutic efficacy in preclinical models. Ultimately, it rigorously evaluates existing constraints and delineates potential and barriers for clinical translation.”
Comments 4: I recommend revising sentences with redundancy (e.g., lines 11–23 in Abstract) for conciseness. Response 4: Thank you for this valuable suggestion. We agree that several sentences in the specified section of the abstract (lines 11–23) were redundant and compromised conciseness. We have carefully revised this section accordingly. The specific revised text is presented in our response to response 3, as this single revision also addresses the issue of precision in the abstract.
Comments 5: Please, can you check all references for formatting consistency? Some in-text citations (e.g., “[10, 11]”) show double brackets or inconsistent style. Response 5: Thank you for your careful review and valuable feedback. We sincerely apologize for the inconsistencies in our reference formatting, particularly the in-text citation errors you pointed out (e.g., “[10, 11]”). We have now conducted a thorough check and revision of all in-text citations and the entire reference list to ensure they fully comply with the journal's formatting guidelines. All inconsistencies, including the double-bracket issue, have been corrected.
Comments 6: I suggest you critically evaluate whether enough independent repetitions and statistical analyses were available in cited experimental articles (especially where western blots or animal studies are referenced). Response 6: Thank you for this insightful and constructive suggestion. We completely agree that a critical perspective on the experimental rigor of cited literature is essential for a high-quality review. We wish to clarify that throughout the preparation of this manuscript, we have made our best effort to prioritize citing studies from high-impact, peer-reviewed journals. It is our understanding that these studies generally adhere to rigorous experimental design, for instance, by providing a sufficient number of independent repetitions and appropriate statistical analyses for data from Western blots and animal studies. We have strived to present these findings in a balanced manner. Furthermore, to underscore the importance of the point you have raised, we have now added a commentary in our "Limitations and Future Perspectives" sectionIn this new passage, we explicitly call for future research to place a greater emphasis on methodological rigor, including ensuring adequate biological replicates and transparent statistical reporting, to enhance the reliability and reproducibility of findings. Details are listed as follows. (5. Conclusion, Current Limitations, and Future Perspectives, Lines 664-677) “In comprehensively evaluating the current literature on natural product-induced ferroptosis, maintaining a critical perspective on the methodological rigor of preclinical studies is essential. Although existing research has provided valuable insights, some studies, particularly early exploratory work, may have limitations in data presentation and statistical analysis. For instance, many conclusions relying on Western Blots sometimes lack sufficient quantification and an adequate number of biological replicates to support their reliability. Similarly, in some animal studies, small sample sizes (n-numbers) may limit their statistical power, making the generalizability of the conclusions questionable. Moreover, many findings are highly dependent on a few specific cancer cell lines, which may not fully reflect the high degree of heterogeneity in breast cancer among actual patients. Therefore, future research must, while pursuing mechanistic innovation, place greater emphasis on adhering to rigorous experimental standards to ensure the reliability and reproducibility of results, thereby laying a more solid foundation for future clinical translation.”
Comments 7: Please, can you strengthen the section on limitations (Conclusion, lines 584–659)? It reads more like a perspective than a critical review of gaps in the field. Response 7: Thank you for this insightful feedback. We agree that our original "Limitations" section focused more on future perspectives and lacked a sufficiently critical analysis of the current gaps in the field. To address this, we have substantially revised and restructured this section (now Section 5.1, titled "A Critical Review of Gaps and Limitations") to provide a rigorous critique of the key challenges. Our revision involved a deeper literature review to identify and synthesize these limitations more robustly. We have reframed the discussion to focus on several critical gaps, including: (1) the mechanistic complexity arising from the multi-target nature of natural products; (2) fundamental pharmacokinetic barriers, such as poor bioavailability; (3) the methodological limitations of many preclinical studies; (4) the over-reliance on simplified experimental models that fail to capture tumor heterogeneity; (5) the critical lack of validated biomarkers for clinical trial design; and (6) the potential for on-target toxicity in non-cancer tissues. Details are listed as follows. ( 5.1 A Critical Review of Gaps and Limitations, Lines 638-725) " A core challenge of natural products lies in their inherent multi-target pharmaco-logical properties. While this characteristic may offer potential advantages for synergistic therapy, it also introduces complexity to the traditional, single-target-based drug discovery paradigm [172, 191]. For instance, the proferroptotic effects of curcumin are linked to multiple signaling pathways and broad pro-oxidant activities, making the precise identification of a dominant mechanism highly challenging [192]. To address this, Network Pharmacology offers a powerful solution. By integrating data from medicinal chemistry, genomics, and bioinformatics, researchers can construct complex protein-protein interaction (PPI) networks and analyze them within the context of specific diseases like breast cancer [193]. Subsequently, topological analysis of this network can identify key hub genes with the highest connectivity in regulating ferroptosis [194]. This process transforms a once-vague multi-target problem into a testable scientific hypothesis focused on a few high-priority molecules [127], providing both a macroscopic and precise guide for subsequent mechanistic research and drug optimization [195]. Beyond mechanistic uncertainty, the clinical development of natural products is fundamentally challenged by their poor pharmacokinetic properties. Many lead com-pounds exhibit extremely low oral bioavailability, preventing them from reaching therapeutically relevant concentrations in the blood. For example, although curcumin displays potent anti-cancer activity in vitro, its clinical application is severely limited by a bioavailability often reported to be less than 1%, primarily due to extensive first-pass metabolism [196]. Resveratrol is another classic example of this process; its rapid glucuronidation and sulfation in the liver lead to similarly low bioavailability and a very short plasma half-life, making it impossible to maintain stable therapeutic concentrations via oral administration [197, 198]. Furthermore, the use of non-standardized plant extracts introduces significant batch-to-batch variability, undermining research reproducibility and posing a major regulatory hurdle for clinical translation [199]. In comprehensively evaluating the current literature on natural product-induced ferroptosis, maintaining a critical perspective on the methodological rigor of preclinical studies is essential. Although existing research has provided valuable insights, some studies, particularly early exploratory work, may have limitations in data presentation and statistical analysis. For instance, many conclusions relying on Western Blots sometimes lack sufficient quantification and an adequate number of biological replicates to support their reliability. Similarly, in some animal studies, small sample sizes (n-numbers) may limit their statistical power, making the generalizability of the con-clusions questionable. Moreover, many findings are highly dependent on a few specific cancer cell lines, which may not fully reflect the high degree of heterogeneity in breast cancer among actual patients. Therefore, future research must, while pursuing mechanistic innovation, place greater emphasis on adhering to rigorous experimental standards to ensure the reliability and reproducibility of results, thereby laying a more solid foundation for future clinical translation. It is important to note that, to date, no clinical trials have been specifically designed to evaluate natural products as ferroptosis modulators in breast cancer, and the current body of evidence remains exclusively at the preclinical stage. The reliance on oversimplified preclinical models is another major obstacle hindering translational success. The vast majority of studies are conducted on two-dimensional (2D) cell monolayers, a system that fails to replicate the complex three-dimensional architecture and cell-cell interactions of solid tumors, nor can it accurately assess drug resistance, phenomena better captured by 3D spheroids and organoid systems[200, 201]. More importantly, these models completely ignore the influence of the tumor microenvironment (TME), which has been recognized as a key regulator of ferroptosis[202]. Studies have shown that stromal cells (such as cancer-associated fibroblasts) can actively confer ferroptosis resistance to neighboring cancer cells, a critical biological interaction completely absent in standard in vitro experiments[203]. Finally, these models fail to account for intra-tumoral heterogeneity. A multi-omics study has revealed distinct and heterogeneous ferroptosis phenotypes within triple-negative breast cancer, suggesting that a drug effective against one cell subpopulation may be ineffective against others within the same tumor[48]. This oversimplification likely leads to a systematic overestimation of candidate drug efficacy. This translational gap is further exacerbated by the lack of validated biomarkers for patient stratification and treatment monitoring. Current development of predictive biomarkers has mainly produced computational gene signatures (such as FERscore) based on retrospective dataset analysis, but they lack the prospective clinical validation necessary to guide patient selection[204, 205]. A second major gap is the absence of pharmacodynamic (PD) biomarkers; there are currently no established non-invasive methods (such as specific imaging agents) to confirm whether a therapeutic agent has engaged its target and initiated ferroptosis within a patient's tumor[206]. Finally, the mechanisms regulating resistance are poorly understood, and there are no biomarkers to identify patients with primary resistance or to monitor the emergence of resistant subclones[72, 207]. This series of deficiencies in predictive, pharmacodynamic, and resistance biomarkers makes the rational design of clinical trials impossible and constitutes a fundamental barrier to the progress of this therapeutic strategy. However, the most fundamental limitation of targeting ferroptosis lies in the potential for on-target toxicity in non-cancer tissues. This risk stems from the fact that ferroptosis is a basic physiological process, not a tumor-specific pathway; therefore, the same mechanisms activated in cancer cells can also disrupt homeostasis in normal tissues. The heart is one of the most vulnerable high-risk organs, as its energy metabolism is highly dependent on mitochondrial iron-sulfur clusters, the destabilization of which directly leads to functional impairment. This vulnerability is exemplified by the clinical cardiotoxicity of doxorubicin: a study by Liao et al. showed that the drug triggers ferroptosis by promoting lipid peroxidation in cardiomyocytes[208], while Cantrell et al. further confirmed that the protective effects of ferroptosis inhibitors rely on the direct stabilization of these mitochondrial iron-sulfur clusters[209]. The kidneys also exhibit significant susceptibility, as their renal tubular epithelial cells are rich in polyunsaturated fatty acids (PUFAs), making them prime substrates for lipid peroxidation[210], and inhibiting the key enzyme ACSL4 has been shown to effectively mitigate ferroptotic damage in the kidneys[211]. Furthermore, the central nervous system is highly sensitive to ferroptosis imbalance due to its high iron content. Studies in neurodegenerative disease models have revealed that GPX4 dysfunction is a key determinant of neuronal ferroptosis[32], and pharmacological inhibition of this pathway can alleviate neurotoxicity[212].” References: “32. Dar, Nawab John, Urmilla John, Nargis Bano, Sameera Khan, and Shahnawaz Ali %J Molecular neurobiology Bhat. "Oxytosis/Ferroptosis in Neurodegeneration: The Underlying Role of Master Regulator Glutathione Peroxidase 4 (Gpx4)." 61, no. 3 (2024): 1507-26. 48. Yang, Fan, Yi Xiao, Jia-Han Ding, Xi Jin, Ding Ma, Da-Qiang Li, Jin-Xiu Shi, Wei Huang, Yi-Ping Wang, and Yi-Zhou %J Cell metabolism Jiang. "Ferroptosis Heterogeneity in Triple-Negative Breast Cancer Reveals an Innovative Immunotherapy Combination Strategy." 35, no. 1 (2023): 84-100. e8. 72. Park, Soon Young, Kang Jin Jeong, Alfonso Poire, Dong Zhang, Yiu Huen Tsang, Aurora S Blucher, Gordon B %J Cell Death Mills, and Disease. "Irreversible Her2 Inhibitors Overcome Resistance to the Rsl3 Ferroptosis Inducer in Non-Her2 Amplified Luminal Breast Cancer." 14, no. 8 (2023): 532. 127. Zhang, Yuan, Chenghao Yu, Cheng Peng, and Fu %J International Journal of Biological Sciences Peng. "Potential Roles and Mechanisms of Curcumin and Its Derivatives in the Regulation of Ferroptosis." 20, no. 12 (2024): 4838. 172. Newman, David J, and Gordon M %J Journal of natural products Cragg. "Natural Products as Sources of New Drugs over the Nearly Four Decades from 01/1981 to 09/2019." 83, no. 3 (2020): 770-803. 191. Kabir, Abbas, and Aaron %J Pharmacological research Muth. "Polypharmacology: The Science of Multi-Targeting Molecules." 176 (2022): 106055. 192. Firouzjaei, Ali Ahmadizad, Seyed Hamid Aghaee‐Bakhtiari, Ali Tafti, Kazem Sharifi, Mohammad Hassan Jafari Najaf Abadi, Samaneh Rezaei, Samira %J Cell Biochemistry Mohammadi‐Yeganeh, and Function. "Impact of Curcumin on Ferroptosis‐Related Genes in Colorectal Cancer: Insights from in‐Silico and in‐Vitro Studies." 41, no. 8 (2023): 1488-502. 193. Zhang, Rui, Yuqin Ma, Jing Zhang, Aiguo Meng, and Chunyan %J Journal of Functional Foods Liu. "Salidroside Induces Ferroptosis in Breast Cancer Cells and Enhances the Anticancer Effect of Oxaliplatin." 130 (2025): 106932. 194. Cai, Banglan, Manman Qi, Xue Zhang, Denghai %J Drug Design Zhang, Development, and Therapy. "Integrating Network Pharmacology with in Vitro Experiments to Validate the Efficacy of Celastrol against Hepatocellular Carcinoma through Ferroptosis." (2024): 3121-41. 195. Wang, Longyan, Huiming Huang, Xingxing Li, Lishan Ouyang, Xuejiao Wei, Jinxin Xie, Dongxiao Liu, Peng Tan, and Zhongdong %J Chinese Medicine Hu. "A Review on the Research Progress of Traditional Chinese Medicine with Anti-Cancer Effect Targeting Ferroptosis." 18, no. 1 (2023): 132. 196. Tabanelli, Rita, Simone Brogi, and Vincenzo %J Pharmaceutics Calderone. "Improving Curcumin Bioavailability: Current Strategies and Future Perspectives." 13, no. 10 (2021): 1715. 197. Li, Yingchao, Ran Zhang, Qi Zhang, Meiling Luo, Farong Lu, Zhonggui He, Qikun Jiang, Tianhong %J Journal of Agricultural Zhang, and Food Chemistry. "Dual Strategy for Improving the Oral Bioavailability of Resveratrol: Enhancing Water Solubility and Inhibiting Glucuronidation." 69, no. 32 (2021): 9249-58. 198. Zhang, Jing, Yongya Wu, Yanhong Li, Shutong Li, Jiaxi Liu, Xiao Yang, Guiyang Xia, and Guan %J Phytomedicine Wang. "Natural Products and Derivatives for Breast Cancer Treatment: From Drug Discovery to Molecular Mechanism." 129 (2024): 155600. 199. Aljabali, Alaa AA, Mohammad A Obeid, Rasha M Bashatwah, Esam Qnais, Omar Gammoh, Abdelrahim Alqudah, Vijay Mishra, Yachana Mishra, Mohammad Ahmed Khan, Suhel %J Chemistry Parvez, and Biodiversity. "Phytochemicals in Cancer Therapy: A Structured Review of Mechanisms, Challenges, and Progress in Personalized Treatment." (2025): e202402479. 200. Carvalho, Sandhra M, Alexandra AP Mansur, Izabela B da Silveira, Thaisa FS Pires, Henrique FV Victória, Klaus Krambrock, M Fátima Leite, and Herman S %J Pharmaceutics Mansur. "Nanozymes with Peroxidase-Like Activity for Ferroptosis-Driven Biocatalytic Nanotherapeutics of Glioblastoma Cancer: 2d and 3d Spheroids Models." 15, no. 6 (2023): 1702. 201. Ye, Lei, Fei Zhong, Shishen Sun, Xiaowei Ou, Jie Yuan, Jintao Zhu, Zhiqiang %J Journal of Cancer Research Zeng, and Therapeutics. "Tamoxifen Induces Ferroptosis in Mcf-7 Organoid." 19, no. 6 (2023): 1627-35. 202. Lei, Guang, Li Zhuang, and Boyi %J Cancer cell Gan. "The Roles of Ferroptosis in Cancer: Tumor Suppression, Tumor Microenvironment, and Therapeutic Interventions." 42, no. 4 (2024): 513-34. 203. Liu, Yongcan, Shanchun Chen, Xueying Wan, Rui Wang, Haojun Luo, Chao Chang, Peijin Dai, Yubi Gan, Yuetong Guo, and Yixuan %J Cancer Communications Hou. "Tryptophan 2, 3‐Dioxygenase‐Positive Matrix Fibroblasts Fuel Breast Cancer Lung Metastasis Via Kynurenine‐Mediated Ferroptosis Resistance of Metastatic Cells and T Cell Dysfunction." 44, no. 11 (2024): 1261-86. 204. Hu, Kaimin, Jili Qiu, Yue Hu, Yanyan Wang, Chengcheng Yu, and Yinan %J NPJ Breast Cancer Wu. "Efficacy of Ferscore in Predicting Sensitivity to Ferroptosis Inducers in Breast Cancer." 10, no. 1 (2024): 74. 205. Imam, Murshid, Jiale Ji, Zhijie Zhang, and Shunchao %J Frontiers in Pharmacology Yan. "Targeting the Initiator to Activate Both Ferroptosis and Cuproptosis for Breast Cancer Treatment: Progress and Possibility for Clinical Application." 15 (2025): 1493188. 206. Cao, Pham Hong Anh, Abishai Dominic, Fabiola Ester Lujan, Sanjanaa Senthilkumar, Pratip K Bhattacharya, Daniel E Frigo, and Elavarasan %J Nature Reviews Urology Subramani. "Unlocking Ferroptosis in Prostate Cancer—the Road to Novel Therapies and Imaging Markers." 21, no. 10 (2024): 615-37. 207. Wang, Yuan, Guifang Yu, and Xin %J Cancer Drug Resistance Chen. "Mechanism of Ferroptosis Resistance in Cancer Cells." 7 (2024): 47. 208. Liao, Hai-Han, Wen Ding, Nan Zhang, Zi-Ying Zhou, Zheng Ling, Wen-Jing Li, Si Chen, Qi-Zhu %J Free Radical Biology Tang, and Medicine. "Activation of Ampkα2 Attenuated Doxorubicin-Induced Cardiotoxicity Via Inhibiting Lipid Peroxidation Associated Ferroptosis." 205 (2023): 275-90. 209. Cantrell, Aubrey C, Jessie Besanson, Quinesha Williams, Ngoc Hoang, Kristin Edwards, G Reid Bishop, Yingjie Chen, Heng Zeng, Jian-Xiong %J Journal of molecular Chen, and cellular cardiology. "Ferrostatin-1 Specifically Targets Mitochondrial Iron-Sulfur Clusters and Aconitase to Improve Cardiac Function in Sirtuin 3 Cardiomyocyte Knockout Mice." 192 (2024): 36-47. 210. Bayır, Hülya, Scott J Dixon, Yulia Y Tyurina, John A Kellum, and Valerian E %J Nature Reviews Nephrology Kagan. "Ferroptotic Mechanisms and Therapeutic Targeting of Iron Metabolism and Lipid Peroxidation in the Kidney." 19, no. 5 (2023): 315-36. 211. Dai, Yue, Yuting Chen, Dexiameng Mo, Rui Jin, Yi Huang, Le Zhang, Cuntai Zhang, Hongyu Gao, and Qi %J Communications Biology Yan. "Inhibition of Acsl4 Ameliorates Tubular Ferroptotic Cell Death and Protects against Fibrotic Kidney Disease." 6, no. 1 (2023): 907. 212. Sripetchwandee, Jirapas, Aphisek Kongkaew, Sirinart Kumfu, Titikorn Chunchai, Nipon Chattipakorn, and Siriporn C %J Life Sciences Chattipakorn. "Ferrostatin-1 and Z-Vad-Fmk Potentially Attenuated Iron-Mediated Neurotoxicity and Rescued Cognitive Function in Iron-Overloaded Rats." 313 (2023): 121269.”
Comments 8: I recommend you provide a comparison table of clinical trial data (if any exist) for natural ferroptosis modulators; otherwise, clearly state that evidence is preclinical only. Response 8: Thank you for this crucial point regarding clinical evidence. We agree that this clarification is essential. First, we conducted a systematic search on ClinicalTrials.gov using keywords including ("Breast Cancer" AND "Ferroptosis") and ("Breast Cancer" AND "Natural Product"). Furthermore, we performed supplementary searches on PubMed and Google Scholar. The search results confirm that no clinical trials have been specifically designed to evaluate natural products as ferroptosis modulators in breast cancer to date. To make this explicit, we have added a definitive statement in Section 5. Details are listed as follows. ( 5.1 A Critical Review of Gaps and Limitations, Lines 678-680) "It is important to note that, to date, no clinical trials have been specifically designed to evaluate natural products as ferroptosis modulators in breast cancer, and the current body of evidence remains exclusively at the preclinical stage."
Comments 9: Please, can you ensure figures include evidence of reproducibility (e.g., whether cited experiments used ≥3 independent repeats)? This is critical for scientific rigor. Response 9: Thank you for your valuable suggestion. We completely agree that ensuring the rigor of the evidence supporting the schematic figures in our review is essential for a high-quality manuscript. Following your suggestion, we have conducted a systematic re-examination of the key references that form the basis for all schematic figures summarizing molecular pathways in our manuscript. We wish to clarify that these figures are intended to summarize well-established and validated molecular pathways in the field. Our re-examination confirms that the key cited studies are supported by multiple independent experimental repetitions. Furthermore, we have strived to rely on the most authoritative and high-impact literature in the field to ensure scientific accuracy.
Comments 10: I suggest revising “ferroptosis effectively bypasses therapeutic resistance” (line 20) to a more cautious wording, as most data are preclinical. Response 10: Thank you for this crucial suggestion. We agree that our original wording was too definitive, given that the majority of the data is preclinical. We have accepted your recommendation and revised this sentence, along with other similar statements throughout the manuscript, to adopt more cautious and scientific language. Details are listed as follows. (Abstract, Lines 15-16) “Ferroptosis, an iron-dependent form of regulated cell death, is emerging as a novel strategy to overcome these obstacles.”
Comments 11: I recommend including a critical discussion of pharmacokinetic challenges (bioavailability, solubility) earlier in the review, not only in the conclusion. Response 11: Thank you for this valuable suggestion. We agree that discussing the pharmacokinetic challenges of natural products earlier in the manuscript is crucial. Accordingly, we have introduced a new paragraph at the beginning of Section 4. This paragraph now systematically addresses the common pharmacokinetic hurdles, such as poor bioavailability, thereby providing the necessary critical context for evaluating the preclinical data of the compounds discussed thereafter. We believe this revision provides a more balanced and comprehensive perspective. Details are listed as follows. (4. Natural Products as Modulators of Ferroptosis in Breast Cancer, Lines 334-343) “However, before delving into the specific biological activities of these various natural compounds, it is crucial to first acknowledge a prevalent and critical challenge: their common pharmacokinetic deficiencies. Many promising natural products, including highly-studied molecules like curcumin and resveratrol, are hampered by issues such as low aqueous solubility, poor intestinal absorption, and rapid in vivo metabolism. These factors collectively lead to low bioavailability, making it extremely difficult to achieve and maintain effective therapeutic concentrations at the tumor site. Therefore, understanding this fundamental bottleneck is essential for objectively evaluating the preclinical data discussed hereafter and for recognizing the necessity of developing optimization strategies, such as advanced delivery systems.”
Comments 12: Please, can you provide clearer separation between compounds with strong mechanistic evidence versus those with only preliminary reports? Response 12: Thank you for this insightful suggestion, which prompted us to perform a rigorous re-evaluation of the evidence supporting each compound discussed in our manuscript. We agree that a clear separation is essential. To achieve this, we first established a clear framework for classifying the evidence. We systematically reviewed each compound to determine whether the evidence was: (a) based on direct experimental validation (in vitro or in vivo) versus review-based citations; and (b) derived directly from breast cancer models versus being extrapolated from other cancer types. Our re-assessment confirmed that the vast majority of compounds cited in our review are supported by direct experimental validation. However, we identified a key area for clarification: for certain classes like Quinones, the primary mechanistic evidence is indeed extrapolated from other cancers. We have now made this explicit in the text to ensure full transparency. (4.5 Other Chemical Classes, Lines 602-612) " Quinones, capable of inducing lipid peroxidation through redox cycling, are pivotal in ferroptosis regulation [178]. While research on breast cancer is scarce, pathways identified in other cancers provide valuable insights. Plumbagin and juglone promote ferroptosis by blocking GPX4 [179], while β‑lapachone enhances the labile iron pool through NCOA4‑mediated ferritinophagy to initiate ferroptosis in colorectal cancer [180]. In breast cancer specifically, de Carvalho et al. showed that bromonaph-thoquinone combined with tannic acid yields synergistic antitumor effects in TNBC cells [181]. Significantly, quinone activity depends largely on NAD(P)H: quinone oxi-doreductase‑1 (NQO1) expression. Wang et al. discovered that tanshinone can inhibit ferroptosis in NQO1‑high cells [182], suggesting that NQO1 status may determine whether quinones function as pro‑death or protective agents in breast cancer [183]." Second, to provide a clear, at-a-glance guide, we have added a new column titled "Strength of Evidence" to all relevant summary tables. We defined "Strong" as evidence from direct experimental studies in breast cancer models, while "Moderate" is used for evidence extrapolated from other cancer types. We are confident that this dual approach—clarifying the evidence source in the text and adding a formal classification in the tables—fully addresses your concern and significantly enhances the manuscript's scientific rigor.
References: “178. Sui, Xinyue, Jichao Wang, Zhiqiang Zhao, Bin Liu, Miaomiao Liu, Min Liu, Cong Shi, Xinjun Feng, Yingxin Fu, and Dayong %J Communications Biology Shi. "Phenolic Compounds Induce Ferroptosis-Like Death by Promoting Hydroxyl Radical Generation in the Fenton Reaction." 7, no. 1 (2024): 199. 179. Zhan, Sheng, Li Lu, Shu-shan Pan, Xiao-qian Wei, Rong-rong Miao, Xiao-hui Liu, Ming Xue, Xiu-kun Lin, and Huan-li %J British journal of cancer Xu. "Targeting Nqo1/Gpx4-Mediated Ferroptosis by Plumbagin Suppresses in Vitro and in Vivo Glioma Growth." 127, no. 2 (2022): 364-76. 180. Zhao, Lei, Hui Miao, Mingqi Quan, Shuhao Wang, Yu Zhang, Houkun Zhou, Xianglan Zhang, Zhenhua Lin, and Junjie %J Chemico-biological interactions Piao. "Β-Lapachone Induces Ferroptosis of Colorectal Cancer Cells Via Ncoa4-Mediated Ferritinophagy by Activating Jnk Pathway." 389 (2024): 110866. 181. de Carvalho, Emanuelle Pangoni, Adriano de Souza Pessoa, Flávia Godoy Iano, Laura Ribeiro, Bianca Leme, Luis Francisco Borges, Mariana Liessa Rovis Sanches, Valdecir Farias Ximenes, Rodrigo Cardoso %J The International Journal of Biochemistry de Oliveira, and Cell Biology. "Antitumor Effect of Bromo-Naphthoquinone Associated with Tannic Acid in Triple Negative Breast Cancer Cells." 177 (2024): 106697. 182. Wang, Tian-Xiang, Kun-Long Duan, Zi-Xuan Huang, Zi-An Xue, Jun-Yun Liang, Yongjun Dang, Ao Zhang, Yue Xiong, Chunyong Ding, and Kun-Liang %J Life science alliance Guan. "Tanshinone Functions as a Coenzyme That Confers Gain of Function of Nqo1 to Suppress Ferroptosis." 6, no. 1 (2023). 183. Yu, Jie, Bingling Zhong, Lin Zhao, Ying Hou, Nana Ai, Jin-Jian Lu, Wei Ge, and Xiuping %J Drug Resistance Updates Chen. "Fighting Drug-Resistant Lung Cancer by Induction of Nad (P) H: Quinone Oxidoreductase 1 (Nqo1)-Mediated Ferroptosis." 70 (2023): 100977.”
Comments 13: I suggest rephrasing “breast cancer is among the most prevalent malignant tumors affecting women worldwide” (line 37) to avoid redundancy with the Abstract. Response 13: Thank you for your valuable suggestion. We agree that the opening sentence of the Introduction was redundant with the Abstract. To address this, we have rephrased the sentence, ensuring it is now distinct in both wording and focus from the statement in the Abstract. Details are listed as follows. (Introduction, Lines 33-34) “Breast cancer represents a major global health burden, continuing to be the most frequently diagnosed malignancy in women.”
Comments 14: Please, can you update references to ensure inclusion of the most recent studies (2023– 2025), especially for clinical translation and nanodelivery strategies? Response 14: Thank you for highlighting the importance of including the most recent studies. We agree that this is crucial for the timeliness and relevance of our review. Following your suggestion, we have conducted a comprehensive literature search using PubMed and Google Scholar, with a time frame limited to publications from 2023 to the present. As a result, we have incorporated numerous recent references throughout the manuscript, with a particular focus on supplementing the sections on nanodelivery strategies and clinical translation. Details are listed as follows. (5.2 Future Perspectives and Strategies to Surmount Obstacles, Lines 740-752, Lines 765-780) " Innovative drug delivery technologies are crucial for translating the potential of natural ferroptosis inducers into clinical reality, with developments extending far beyond traditional liposomes and polymer micelles. Recent advances have focused on sophisticated biomimetic nanoplatforms, which are camouflaged with membranes de-rived from macrophages or cancer cells to evade immune clearance and achieve superior tumor-homing capabilities [218-220]. Beyond passive targeting, these nanotechnologies are also engineered to actively remodel the tumor microenvironment (TME) to amplify the ferroptotic effect. This can be achieved through various strategies, such as depleting intracellular glutathione (GSH) to dismantle the cell's primary antioxidant defense system [221, 222], or delivering iron ions to catalyze the Fenton reaction, ensuring a continuous supply of cytotoxic reactive oxygen species [223]. Furthermore, smart designs like size-switchable nanocapsules are being developed to overcome physical barriers and enhance deep tumor penetration [224, 225]."
" Finally, future research must shift its focus from in vitro cell experiments to more clinically relevant models to bridge the current translational gap. A priority research direction is the validation of promising natural products in appropriate preclinical animal models. For example, Tanshinone IIA, which has shown potent ferroptosis-inducing activity in triple-negative breast cancer (TNBC) cells, could be administered via tail vein injection in mice bearing 4T1 orthotopic tumors [15]. Such a study should not only evaluate its efficacy in inhibiting primary tumor growth and lung metastasis but also confirm its in vivo ferroptosis-inducing mechanism by detecting levels of 4-HNE (a marker of lipid peroxidation) in tumor tissues [228]. Concurrently, con-ducting translational research based on human clinical samples is also crucial. A specific and feasible experiment would be to use tumor tissue microarrays (TMAs) containing samples from hundreds of breast cancer patients to quantitatively assess the expression level of the key negative ferroptosis regulator, GPX4, using immunohistochemistry (IHC) [104]. Subsequently, Kaplan-Meier analysis could be used to test whether low GPX4 expression is significantly associated with longer disease-free survival (DFS) in patients, particularly those receiving anthracycline-based chemotherapy[55, 229]." References: “15. Ge, Anqi, Qi He, Da Zhao, Yuwei Li, Junpeng Chen, Ying Deng, Wang Xiang, Hongqiao Fan, Shiting Wu, Yan %J Journal of Cellular Li, and Molecular Medicine. "Mechanism of Ferroptosis in Breast Cancer and Research Progress of Natural Compounds Regulating Ferroptosis." 28, no. 1 (2024): e18044. 55. Sha, Rui, Yaqian Xu, Chenwei Yuan, Xiaonan Sheng, Ziping Wu, Jing Peng, Yaohui Wang, Yanping Lin, Liheng Zhou, and Shuguang %J EBioMedicine Xu. "Predictive and Prognostic Impact of Ferroptosis-Related Genes Acsl4 and Gpx4 on Breast Cancer Treated with Neoadjuvant Chemotherapy." 71 (2021). 104. Gong, Rong, Xiaoya Wan, Shilong Jiang, Yidi Guan, Yizhi Li, Ting Jiang, Zonglin Chen, Changxin Zhong, Linhao He, Zhongyuan %J Cell Death Xiang, and Differentiation. "Gpx4-Autac Induces Ferroptosis in Breast Cancer by Promoting the Selective Autophagic Degradation of Gpx4 Mediated by Traf6-P62." (2025): 1-16. 218. Cao, Zhengcong, Xiao Liu, Wangqian Zhang, Keying Zhang, Luxiang Pan, Maorong Zhu, Haozhe Qin, Cheng Zou, Weizhong Wang, and Cong %J Acs Nano Zhang. "Biomimetic Macrophage Membrane-Camouflaged Nanoparticles Induce Ferroptosis by Promoting Mitochondrial Damage in Glioblastoma." 17, no. 23 (2023): 23746-60. 219. Mu, Yeteng, Yuxin Fan, Lianping He, Nannan Hu, Han Xue, Xingang Guan, and Zhijian %J Cancer Nanotechnology Zheng. "Enhanced Cancer Immunotherapy through Synergistic Ferroptosis and Immune Checkpoint Blockade Using Cell Membrane-Coated Nanoparticles." 14, no. 1 (2023): 83. 220. Li, Yu, Hongji Li, Keying Zhang, Chao Xu, Jingwei Wang, Zeyu Li, Yike Zhou, Shaojie Liu, Xiaolong Zhao, and Zhengxuan %J Advanced Science Li. "Genetically Engineered Membrane‐Coated Nanoparticles for Enhanced Prostate‐Specific Membrane Antigen Targeting and Ferroptosis Treatment of Castration‐Resistant Prostate Cancer." 11, no. 33 (2024): 2401095. 221. Song, Wen‐Fang, Jin‐Yue Zeng, Ping Ji, Zi‐Yi Han, Yun‐Xia Sun, and Xian‐Zheng %J Small Zhang. "Self‐Assembled Copper‐Based Nanoparticles for Glutathione Activated and Enzymatic Cascade‐Enhanced Ferroptosis and Immunotherapy in Cancer Treatment." 19, no. 35 (2023): 2301148. 222. Zhang, Gaorui, Jiazhi Duan, Feiran Yu, Yafei Qi, Songbo Zhao, Yuxuan Zhao, Xiaoyu Han, Hong Liu, Yuanhua Sang, and Dexin %J Advanced Functional Materials Yu. "Gsh‐Responsive Mn2+ Burst Nanoboxes as Mitophagy Intervention Agents Augment Ferroptosis and Chemoimmunotherapy in Triple‐Negative Breast Cancer." (2025): 2500491. 223. Yang, Jing, Jiaoyang Zhu, Bin Ren, Haobin Cai, Zongheng Li, Qingdeng Fan, Wei Xiong, Jie Feng, Chenggong Yan, and Ge %J Chemical Engineering Journal Wen. "A Hollow Mesoporous Iron Oxide Nanoparticle to Strengthen Fenton Reaction and Weaken Antioxidant Defense Systems for High Efficacy Tumor Ferroptosis Therapy." 497 (2024): 154470. 224. Wang, Jingjing, Zhi Fang, Chenyang Zhao, Zhaoli Sun, Shen Gao, Biao Zhang, Daping Qiu, Meng Yang, Fugeng Sheng, and Song %J Advanced Materials Gao. "Intelligent Size‐Switchable Iron Carbide‐Based Nanocapsules with Cascade Delivery Capacity for Hyperthermia‐Enhanced Deep Tumor Ferroptosis." 36, no. 9 (2024): 2307006. 225. Hu, Ziwei, Haixin Tan, Yicheng Ye, Wenxin Xu, Junbin Gao, Lu Liu, Lishan Zhang, Jiamiao Jiang, Hao Tian, and Fei %J Advanced Materials Peng. "Nir‐Actuated Ferroptosis Nanomotor for Enhanced Tumor Penetration and Therapy." 36, no. 49 (2024): 2412227. 226. Gong, Guowei, Kumar Ganesan, Yaqun Liu, Yongping Huang, Yuting Luo, Xuexu Wang, Zhenxia Zhang, and Yuzhong %J Journal of Ethnopharmacology Zheng. "Danggui Buxue Tang Improves Therapeutic Efficacy of Doxorubicin in Triple Negative Breast Cancer Via Ferroptosis." 323 (2024): 117655. 227. Yuan, Jialin, Cong Liu, Chengwei Jiang, Ning Liu, Zhaoying Yang, and Hua %J Scientific Reports Xing. "Rsl3 Induces Ferroptosis by Activating the Nf-Κb Signalling Pathway to Enhance the Chemosensitivity of Triple-Negative Breast Cancer Cells to Paclitaxel." 15, no. 1 (2025): 1654. 228. Zhong, Tianfei, Ying Li, Meng Jin, Jingqun Liu, Zhenyu Wu, Feiye Zhu, Lisha Zhao, Yongsheng Fan, Li Xu, Jinjun %J Cell Death Ji, and Disease. "Downregulation of 4-Hne and Foxo4 Collaboratively Promotes Nsclc Cell Migration and Tumor Growth." 15, no. 7 (2024): 546. 229. Zhao, Jiazheng, Ning Zhang, Xiaowei Ma, Ming Li, and Helin %J Cell Death Discovery Feng. "The Dual Role of Ferroptosis in Anthracycline-Based Chemotherapy Includes Reducing Resistance and Increasing Toxicity." 9, no. 1 (2023): 184.”
Comments 15: I recommend you unify terminology: sometimes “ferroptosis modulators,” other times “inducers”—please standardize for clarity. Response 15: We are very grateful for this excellent suggestion. You correctly identified our interchangeable use of "ferroptosis modulators" and "ferroptosis inducers", and we apologize for this oversight. Ultimately, we decided to standardize our terminology and consistently use "ferroptosis modulators" throughout the manuscript.
Comments 16: Please, can you revise long sentences (e.g., lines 83–95) that are difficult to follow? Breaking them will improve readability and scientific clarity. Response 16: Thank you for this helpful suggestion. We agree that the long sentences in this section made it difficult to read. We have revised the paragraph by breaking the complex sentences into shorter, more focused ones to enhance readability and scientific clarity. Guided by your feedback, we also reviewed the entire manuscript for similar issues and made additional revisions to improve overall clarity. Details are listed as follows. (Introduction, Lines 77-79) “This study summarizes recent advancements in modulating ferroptosis for breast cancer therapy. Specifically, it focuses on the role of natural products in this process. It emphasizes the fundamental molecular mechanisms. It aggregates preclinical research and assesses treatment efficacy in experimental models. Additionally, it offers a theoretical foundation for subsequent study and pharmaceutical innovation. Future research in this field is anticipated to focus on multiple areas. A key research priority is the development of biomarker-driven patient stratification systems to enable precision therapy. In parallel, it is essential to enhance the pharmacokinetic properties of these compounds through medicinal-chemistry optimization and nanodelivery technologies. Equally important is a deeper analysis of the fundamental interplay between the tumor microenvironment and ferroptosis regulatory networks. It is essential to elucidate the interaction between metabolic reprogramming, immunological regulation, and ferroptosis. Furthermore, establishing standardized methodologies for the detection of ferroptosis is essential. It is also critical to develop dynamic monitoring techniques for ferroptosis-related biomarkers to facilitate clinical translation.”
Comments 17: I suggest you add a section discussing potential adverse effects of ferroptosis induction in non-cancer tissues, which is underrepresented. Response 17: Thank you for this crucial suggestion. We agree that discussing the potential adverse effects of ferroptosis induction in non-cancer tissues is critical. Accordingly, we have added a new paragraph to our "Limitations and Future Perspectives" section This new paragraph first explains the physiological basis for this toxicity. It then details the mechanisms of damage in vulnerable organs like the heart and kidneys, while also referencing findings from the literature on how these specific toxicities can be mitigated. Details are listed as follows. (5. Conclusion, Current Limitations, and Future Perspectives, Lines 708-725) “However, the most fundamental limitation of targeting ferroptosis lies in the po-tential for on-target toxicity in non-cancer tissues. This risk stems from the fact that ferroptosis is a basic physiological process, not a tumor-specific pathway; therefore, the same mechanisms activated in cancer cells can also disrupt homeostasis in normal tissues. The heart is one of the most vulnerable high-risk organs, as its energy metabolism is highly dependent on mitochondrial iron-sulfur clusters, the destabilization of which directly leads to functional impairment. This vulnerability is exemplified by the clinical cardiotoxicity of doxorubicin: a study by Liao et al. showed that the drug trig-gers ferroptosis by promoting lipid peroxidation in cardiomyocytes [208], while Cantrell et al. further confirmed that the protective effects of ferroptosis inhibitors rely on the direct stabilization of these mitochondrial iron-sulfur clusters [209]. The kidneys also exhibit significant susceptibility, as their renal tubular epithelial cells are rich in polyunsaturated fatty acids (PUFAs), making them prime substrates for lipid peroxidation [210], and inhibiting the key enzyme ACSL4 has been shown to effectively mitigate ferroptotic damage in the kidneys [211]. Furthermore, the central nervous system is highly sensitive to ferroptosis imbalance due to its high iron content. Studies in neurodegenerative disease models have revealed that GPX4 dysfunction is a key determinant of neuronal ferroptosis [32], and pharmacological inhibition of this pathway can alleviate neurotoxicity [212].” Refences: “32. Dar, Nawab John, Urmilla John, Nargis Bano, Sameera Khan, and Shahnawaz Ali %J Molecular neurobiology Bhat. "Oxytosis/Ferroptosis in Neurodegeneration: The Underlying Role of Master Regulator Glutathione Peroxidase 4 (Gpx4)." 61, no. 3 (2024): 1507-26. 208. Liao, Hai-Han, Wen Ding, Nan Zhang, Zi-Ying Zhou, Zheng Ling, Wen-Jing Li, Si Chen, Qi-Zhu %J Free Radical Biology Tang, and Medicine. "Activation of Ampkα2 Attenuated Doxorubicin-Induced Cardiotoxicity Via Inhibiting Lipid Peroxidation Associated Ferroptosis." 205 (2023): 275-90. 209. Cantrell, Aubrey C, Jessie Besanson, Quinesha Williams, Ngoc Hoang, Kristin Edwards, G Reid Bishop, Yingjie Chen, Heng Zeng, Jian-Xiong %J Journal of molecular Chen, and cellular cardiology. "Ferrostatin-1 Specifically Targets Mitochondrial Iron-Sulfur Clusters and Aconitase to Improve Cardiac Function in Sirtuin 3 Cardiomyocyte Knockout Mice." 192 (2024): 36-47. 210. Bayır, Hülya, Scott J Dixon, Yulia Y Tyurina, John A Kellum, and Valerian E %J Nature Reviews Nephrology Kagan. "Ferroptotic Mechanisms and Therapeutic Targeting of Iron Metabolism and Lipid Peroxidation in the Kidney." 19, no. 5 (2023): 315-36. 211. Dai, Yue, Yuting Chen, Dexiameng Mo, Rui Jin, Yi Huang, Le Zhang, Cuntai Zhang, Hongyu Gao, and Qi %J Communications Biology Yan. "Inhibition of Acsl4 Ameliorates Tubular Ferroptotic Cell Death and Protects against Fibrotic Kidney Disease." 6, no. 1 (2023): 907. 212. Sripetchwandee, Jirapas, Aphisek Kongkaew, Sirinart Kumfu, Titikorn Chunchai, Nipon Chattipakorn, and Siriporn C %J Life Sciences Chattipakorn. "Ferrostatin-1 and Z-Vad-Fmk Potentially Attenuated Iron-Mediated Neurotoxicity and Rescued Cognitive Function in Iron-Overloaded Rats." 313 (2023): 121269.”
Comments 18: Please, can you check whether your figures are original or adapted from previous work, and if adapted, provide proper attribution? Response 18: Thank you for your valuable suggestion. We completely agree that providing clear and proper credit for all figures is a fundamental requirement of academic publishing. Accordingly, we have reviewed and revised all figures in the manuscript. Figures 1 and 2 are original illustrations created with Figdraw.com, as stated in their legends. For all chemical structure figures (Figures 3-7), we have also explicitly attributed their sources in the respective legends: structures from the PubChem database are now annotated with their unique Compound IDs (CIDs), while those from specific publications are directly cited. For example, the legend for Figure 4 has been updated as follows: " Figure 4. Chemical Structures of Representative Terpenoids and Their Derivatives that Modulate Ferroptosis.
|
Round 2
Reviewer 2 Report
Comments and Suggestions for Authors
After significant revisions, I recommend for publication.